# Deletion of Trim28 in committed adipocytes promotes obesity but preserves glucose tolerance

Simon T. Bond [1,2], Emily J. King [1,2], Darren C. Henstridge [1,2,3], Adrian Tran [1,2], Sarah C. Moody [1], Christine Yang[1], Yingying Liu[1], Natalie A. Mellett[1], Artika P. Nath[1], Michael Inouye[1,4], Elizabeth J. Tarling [5], Thomas Q. de Aguiar Vallim[5], Peter J. Meikle [1], Anna C. Calkin[1,2] & Brian G. Drew [1,2 ✉]

The effective storage of lipids in white adipose tissue (WAT) critically impacts whole body energy homeostasis. Many genes have been implicated in WAT lipid metabolism, including tripartite motif containing 28 (*Trim28*), a gene proposed to primarily influence adiposity via epigenetic mechanisms in embryonic development. However, in the current study we demonstrate that mice with deletion of Trim28 specifically in committed adipocytes, also develop obesity similar to global Trim28 deletion models, highlighting a post-developmental role for Trim28. These effects were exacerbated in female mice, contributing to the growing notion that Trim28 is a sex-specific regulator of obesity. Mechanistically, this phenotype involves alterations in lipolysis and triglyceride metabolism, explained in part by loss of *Klf14* expression, a gene previously demonstrated to modulate adipocyte size and body composition in a sex-specific manner. Thus, these findings provide evidence that Trim28 is a bona fide, sex specific regulator of post-developmental adiposity and WAT function.

[1] Baker Heart & Diabetes Institute, Melbourne, VIC, Australia 3004. [2] Central Clinical School, Monash University, Melbourne, VIC, Australia 3004. [3] College of Health and Medicine, School of Health Sciences, University of Tasmania, Launceston, TAS, Australia. [4] Department of Public Health and Primary Care, University of Cambridge, Cambridge, UK. [5] Department of Medicine, University of California Los Angeles, Los Angeles, CA, USA. ✉email: brian.drew@baker.edu.au

Obesity is increasing in prevalence worldwide and can lead to a range of chronic health complications, including insulin resistance, type 2 diabetes (T2D) and nonalcoholic fatty liver disease. Such complications arise in part from the deposition of lipid in peripheral tissues, such as the liver and muscle that leads to lipotoxicity[1]. This lipotoxicity mostly occurs when adipocytes become saturated (excessively hypertrophic) and incapable of storing additional lipid[2], which affects their metabolic and endocrine functions, and thus results in an unhealthy phenotype. Recent studies have shown that stimulating adipogenesis, or enhancing healthy WAT expansion via hyperplasia, can reduce obesity-induced complications and result in a metabolically healthy phenotype[3]. This is exemplified in humans where obese but metabolically healthy individuals have regular sized adipocytes. In contrast, nonobese individuals whom are metabolically unhealthy have significantly larger adipocytes[4,5]. Thus, the identification of novel pathways and therapeutics that promote healthy WAT expansion may provide benefit for the treatment of obesity and the metabolic syndrome.

Numerous pathways have previously been shown to impact on adipocyte expansion and differentiation, none more so than PPARγ, considered to be the master regulator of adipogenesis and a direct target of the thiazolidinedione class of insulin sensitizers[6,7]. However, as researchers have explored deeper into these critical mechanisms, it has become apparent that many pathways converge to regulate adipocyte function and have been validated as bona fide modulators of obesity. Among the proteins identified, tripartite motif protein 28 (Trim28), also known as KAP1 and Tif1β, has been genetically associated with increased fat mass (FM) in mice and humans[8–10]; however, the mechanisms for this still remain largely undefined.

Trim28 is a transcriptional corepressor that forms part of a complex that interacts with the KRAB-containing zinc finger transcription family of proteins[11]. Trim28 also harbors SUMO E3 ligase activity independent of its gene regulatory actions[11]. Recent studies have revealed that Trim28 plays a key role in the regulation of multiple aspects of mammalian physiology[12]. Furthermore, it has been established through siRNA screens, that Trim28 is associated with autophagy, with Trim28 being shown to regulate microRNAs that target autophagy-related genes and subsequently regulate autophagosome formation[13,14].

With regard to adiposity, it has been demonstrated that mice heterozygous for a global null mutation in the Trim28 gene had a distinct variance in body weight favoring obesity compared with their wild-type littermates[9,10]. These mice had a significant increase in FM and presented with symptoms of metabolic syndrome, including glucose intolerance[10]. In contrast, other groups have demonstrated that these mice did not exhibit features of the metabolic syndrome, reporting that Trim28 haploinsufficiency leads to a metabolically healthy phenotype in both rodents and humans[8]. Mechanistically, this was attributed to exposure of global Trim28 haploinsufficient mice and humans to environmental triggers, which were specifically linked to alterations in an imprinted gene network influenced by reduced Trim28 expression[8]. Interestingly, Dalgaard et al. also provided data to suggest that post-developmental adipocyte-specific deletion of Trim28 did not precipitate the same obese phenotype; however, the data provided to support this was somewhat limited[8]. The authors interpreted these findings as evidence that the obesity phenotype induced by Trim28 haploinsufficiency, was primarily driven by a developmental mechanism and did not occur in models where Trim28 was deleted in more committed tissue-specific cell lineages (i.e., developed adipocytes)[8].

Despite strong evidence from global haploinsufficient models linking Trim28 with adiposity, no studies have rigorously tested the effect of Trim28 deletion in metabolic tissues post-development. Thus, we sought to investigate the effect of Trim28 deletion in committed adipocytes.

Herein, we demonstrate that the deletion of Trim28 in committed adipocytes (Trim28 adi-KO (knock out)) recapitulates the obese phenotype observed in global haploinsufficient rodents and humans, challenging previous suggestions that this effect does not occur in post-developmental tissue. Moreover, we demonstrate that female Trim28 adi-KO mice have an exacerbated obesity phenotype, consistent with the sex-specific effects demonstrated in global models. Finally, we demonstrate that deletion of Trim28 leads to loss of expression of the gene Klf14, a well-described sex-specific metabolic regulator, revealing a previously unexplored link between Trim28, Klf14, and the development of obesity.

## Results

**Analysis of Trim28 expression in adipocytes and generation of mice deleted for Trim28 in committed adipocytes**. The expression of Trim28 during adipocyte development has to our knowledge, not been outlined in detail. Thus, we analyzed Trim28 protein and mRNA expression throughout adipogenesis in cultured 3T3-L1 adipocytes. We reveal that Trim28 expression was high in fibroblast-like precursor cells, but was progressively reduced as cells differentiated (Fig. 1a, b). The lowest level expression was observed at approximately day 6 post differentiation (Fig. 1c), when adipocytes are largely undergoing the final stages of lipid loading. This association is supported by data from a large mouse genetic reference panel (hybrid mouse diversity panel—HMDP), which demonstrated that adipose tissue Trim28 mRNA expression was negatively correlated with indices of FM (Table 1). These findings support a role for Trim28 being regulated in adipocytes during lipid loading, thus we sought to investigate the effects of Trim28 deletion in adipose tissue in vivo.

We generated mice with Trim28 deletion in committed adipocytes (Trim28 adi-KO) by crossing floxed Trim28 (Trim28fl/fl) mice with AdipoQ-Cre (adiponectin-cre) mice. Cre-recombinase driven by the adiponectin promoter is expressed in committed (developed) adipocytes, and thus allows us to study the effect of Trim28 KO in all adipocytes (immature and mature), while not impacting on progenitors and stem-like cells. Immunoblotting of white adipose tissue (WAT) demonstrated that Trim28 was significantly ($p < 0.05$) reduced in male and female KO mice (Fig. 1d, e), with residual expression likely a result of intact expression of Trim28 in non-adipocytes (progenitors, immune cells, and vascular cells). Using quantitative PCR (qPCR), we demonstrated that Trim28 mRNA expression was also reduced in both WAT and brown adipose tissue (BAT), but was not reduced in other tissues, such as skeletal muscle (quadriceps = Quad) and liver (Fig. 1f).

**Trim28 adi-KO mice display increased adiposity but preserved glucose tolerance on a chow diet**. To understand the metabolic status of these mice, we first studied cohorts of both male and female WT (Trim28fl/fl) and Trim28 adi-KO (Trim28fl/fl AdipoQ-Cre) littermates that were fed a chow diet for 24 weeks. Total body weight increased with time over the study period in all mice with no significant difference observed between Trim28 WT and adi-KO mice in either sex (Fig. 2a, b). Male Trim28 adi-KO mice demonstrated a trend for increased FM (percentage of body mass —FM% BW), which began to differ from ~8 weeks of age (Fig. 2c). Absolute mass of WAT and FM to lean mass (LM) ratios demonstrated similar trends (Supplementary Fig. 1C, D). Similarly, female mice exhibited a significant (genotype $p = 0.0034$, genotype × time $p = 0.0215$) increase in FM from ~8 weeks of age, which continued to increase over the duration of the study (Fig. 2d and Supplementary Fig. 1G, H). There was no

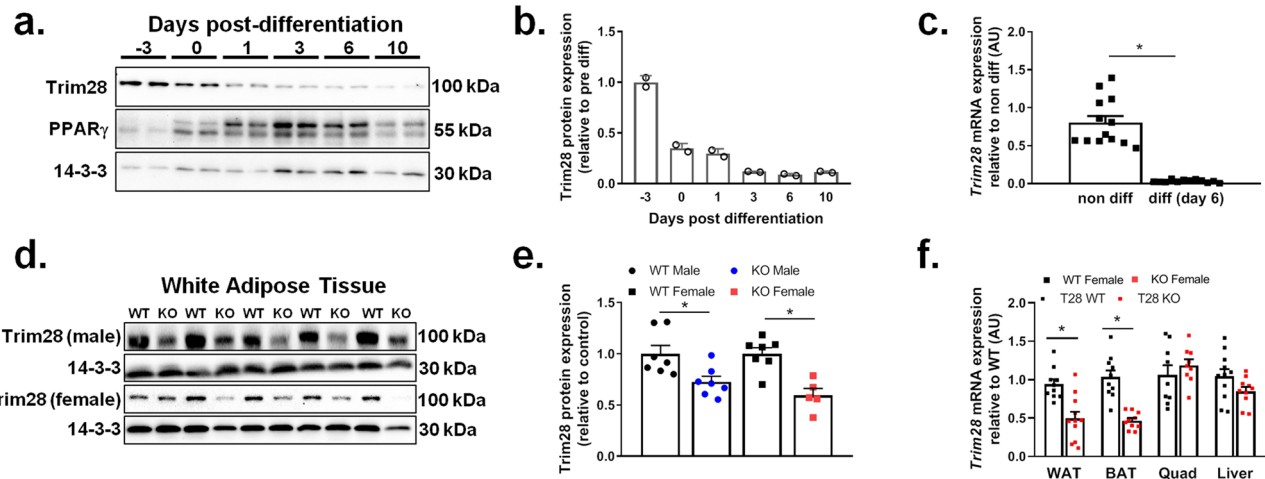

**Fig. 1 Trim28 expression is reduced in differentiated adipocytes and in WAT from mice deleted for Trim28.** Protein and gene analyses were performed in 3T3-L1 adipocytes at different time points pre and post differentiation. **a** Representative immunoblot for Trim28, PPARγ, and 14-3-3 over a differentiation time course in 3T3-L1 cells, and (**b**) densitometry analysis of the Trim28 blots normalized to 14-3-3 and presented as relative to pre-differentiated 3T3-L1 Trim28 expression, $n = 2$. **c** Trim28 mRNA expression in non-differentiated (non-diff) and day 6 differentiated (diff day 6) 3T3-L1 cells, $*p < 0.05$ versus non-diff as analyzed by ANOVA with Fisher's LSD post hoc testing, $n = 13$ non-diff, $n = 12$ diff. Cell culture experiments were repeated at least three times each. Protein and gene expression analyses performed in gonadal fat pads from Trim28 wild type (WT) and adipose-specific Trim28 KO (KO) mice. **d** Immunoblot of Trim28 and 14-3-3 in WT and adi-KO male and female fat pads. **e** Trim28 densitometry normalized to 14-3-3 relative to WT ($n = 7$ WT and adi-KO male mice, $n = 7$ WT and $n = 5$ adi-KO female mice), $*p < 0.05$ versus WT mice as analyzed by ANOVA with Fisher's LSD post hoc testing. **f** Trim28 mRNA expression in WAT, BAT, quad, and liver of WT and adi-KO mice ($n = 10$ per group), $*p < 0.05$ versus WT mice as analyzed by ANOVA with Fisher's LSD post hoc testing, $n = 12$ WT and $n = 12$ KO. All data are presented as mean ± SEM. WAT white adipose tissue, BAT brown adipose tissue, Quad quadriceps muscle.

**Table 1 Phenotypic traits correlated with _Trim28_ expression in mouse adipose tissue from >100 strains of mice from the hybrid mouse diversity panel (HMDP). NMR = nuclear magnetic resonance. Data statistically analyzed using Biweight midcorrelation (Bicor).**

| Phenotype name | Corr. coef. (biweight midcorrelation) | Adjusted P value |
|---|---|---|
| Gonad fat percent | −0.364 | 2.91E−04 |
| Mesenteric fat | −0.363 | 3.00E−04 |
| NMR body fat percentage | −0.361 | 3.31E−04 |
| Phospholipid transfer protein activity | −0.355 | 5.12E−04 |
| Gonad fat | −0.349 | 5.25E−04 |
| Fat mass | −0.346 | 6.01E−04 |
| [Log] body weight | −0.345 | 6.11E−04 |
| Subcutaneous fat percent | −0.344 | 6.37E−04 |
| Mesenteric fat percent | −0.340 | 7.59E−04 |
| NMR total mass | −0.339 | 7.86E−04 |
| Subcutaneous fat | −0.338 | 7.99E−04 |
| Body weight | −0.335 | 9.00E−04 |
| Renal fat percent | −0.323 | 1.40E−03 |
| Renal fat | −0.318 | 1.66E−03 |
| Right kidney mass | −0.272 | 1.84E−02 |
| Serum amyloid A1 | −0.257 | 1.45E−02 |
| Lecithin–cholesterol acyltransferase | −0.256 | 1.50E−02 |
| Bone mass density (femur) | −0.250 | 1.77E−02 |
| Apolipoprotein C2 | −0.247 | 1.88E−02 |
| Insulin | −0.232 | 2.67E−02 |
| Apolipoprotein D | −0.208 | 4.97E−02 |
| Glucose to insulin | 0.310 | 2.81E−03 |

Data statistically analyzed using biweight midcorrelation (Bicor).
NMR nuclear magnetic resonance.

marked differences in LM observed between WT and adi-KO mice (Supplementary Fig. 1A to Supplementary Fig. 1B and Supplementary Fig. 1E to Supplementary Fig. 1F, respectively), although there was a small but significant change in LM as a percentage of body weight in female mice (Supplementary Fig. 1F). However, no changes were observed in absolute LM (Supplementary Fig. 1E), suggesting that the increased FM was confounding the LM data adjusted for body weight.

We next assessed measures of glucose metabolism, including fasting blood glucose and glucose tolerance in both male and female mice at 6 weeks (Fig. 2e–h: basal), 10 weeks (Fig. 2i–l: mid-study), and 22 weeks (Fig. 2m–p: end of study) of age. Fasting blood glucose levels tended to be lower in male Trim28 adi-KO mice at every time point analyzed (Fig. 2e, i, m), which was statistically significant ($p < 0.05$) at the end of the study (Fig. 2m). Female adi-KO mice demonstrated a decrease in fasting blood glucose ($p < 0.05$) only at the end of the study (Fig. 2m). There were no statistically significant differences in glucose tolerance as determined by oral glucose tolerance test (oGTT) at any time point throughout the study for either male or female mice; however, it did appear that female Trim28 adi-KO mice had a trend for improved glucose tolerance at all time points despite having a higher percentage of body fat (Fig. 2g, k, o). This was most notable at study end (Fig. 2o). Furthermore, there were no changes observed in fasting plasma insulin levels (Supplementary Fig. 1I), suggesting that the lack of glucose intolerance in adi-KO mice may be due to increased insulin sensitivity. This finding was supported by insulin signaling analysis in the liver of adi-KO mice, as shown by western blots for Akt phosphorylation at Serine 473 (Supplementary Fig. 1J, K).

**Female Trim28 adi-KO mice develop more pronounced obesity and preservation of glucose tolerance than male mice on a high-fat diet.** Further to the studies above on chow diet, a similar cohort of mice were generated and fed a high-fat diet (HFD; 43% kCal from fat), then subjected to analyses as described above for

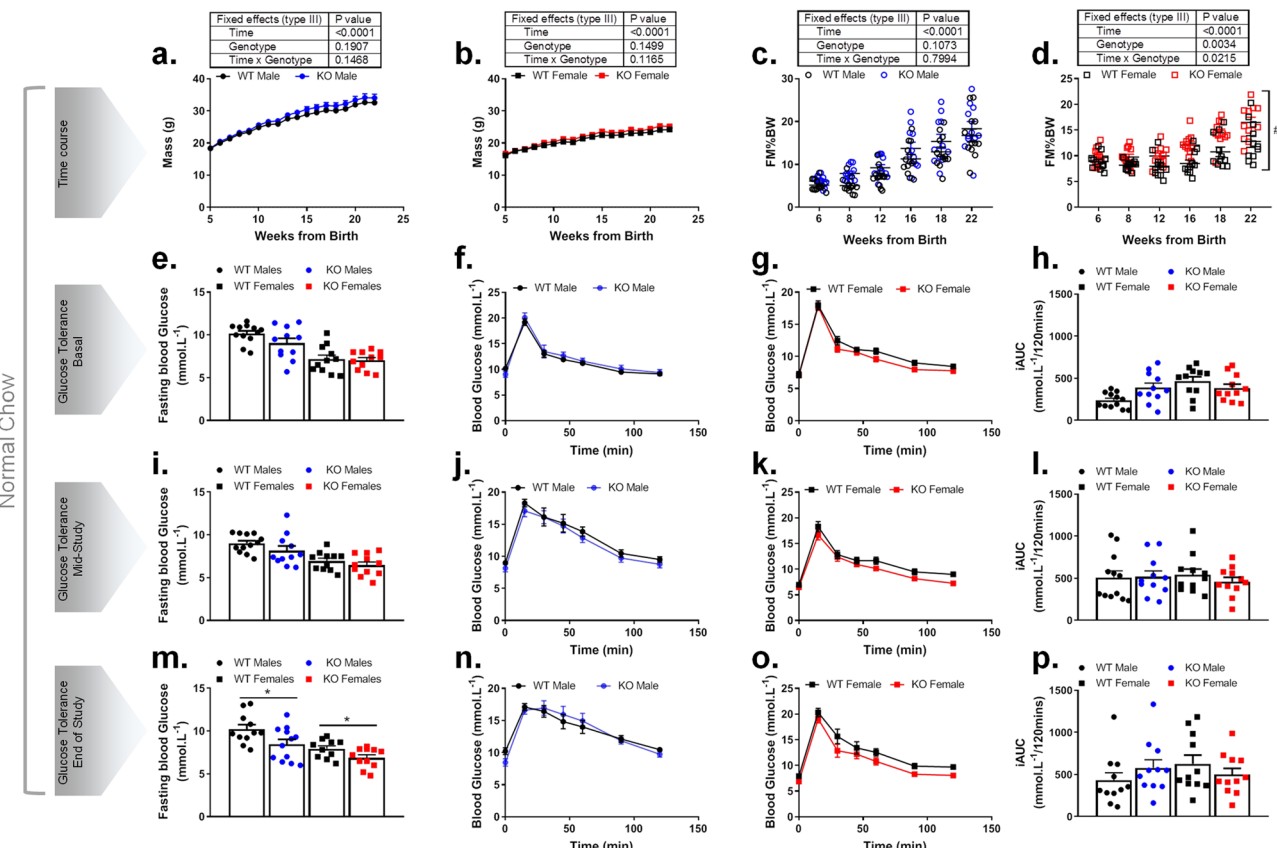

**Fig. 2 Trim28 adi-KO increases adiposity but preserves glucose homeostasis in male and female mice fed a chow diet.** Total body weight of male (**a**) and female (**b**) Trim28 WT and adi-KO (KO) mice over 22-week study period on chow diet. Body composition including body mass and fat mass presented as percent body weight (BW) for (**c**) male mice and (**d**) female mice from 6 weeks of age. Fasting blood glucose and oral glucose tolerance test in male and female mice with corresponding incremental area under the curve (iAUC) at 6 weeks of age (**e–h**; basal), 10 weeks of age (**i–l**; mid-study), and 22 weeks of age (**m–p**; end of study). All data are presented as mean ± SEM, female mice $n = 11$ WT and $n = 11$ KO, and male mice $n = 12$ WT and $n = 11$ KO. All data were analyzed using two-way ANOVA where *$p < 0.05$, or repeated measures mixed-effects model where #$p < 0.05$ for time × genotype interaction.

chow-fed mice. Total body weights of male and female mice increased over the 16 weeks of HFD (Fig. 3a, b), with no significant difference observed between Trim28 WT and adi-KO in male mice (Fig. 3a). However, female Trim28 adi-KO mice were significantly (genotype $p = 0.024$, genotype × time $p < 0.0001$) heavier than WT mice for the duration of the study beginning from ~7 weeks of age (Fig. 3b). Despite no change in total body weight in male mice, FM was significantly (genotype $p = 0.017$, genotype × time $p = 0.025$) increased at baseline, and this effect persisted throughout the entire HFD period (Fig. 3c and Supplementary Fig. 2C, D). Consistent with total body weights, there was a significant (genotype $p = 0.0012$, genotype × time $p < 0.0001$) increase in FM observed in female Trim28 adi-KO mice, which began at ~4 weeks post diet and persisted for the remainder of the study (Fig. 3d and Supplementary Fig. 2G, H). A slight decrease in LM:BW was observed in both male and female Trim28 adi-KO mice over the course of HFD treatment (Supplementary Fig. 2A to Supplementary Fig. 2B and Supplementary Fig. 2E to Supplementary Fig. 2F, respectively); however, as observed with chow diet-fed mice, this effect can be mostly attributed to increases in FM.

In contrast to chow-fed animals, we did not observe any difference in fasting blood glucose between Trim28 WT and adi-KO mice of either sex at any time point following HFD (Fig. 3e, i, m). We also did not observe any difference in glucose tolerance at baseline (Fig. 3e–h; basal) or after 4 weeks of HFD (Fig. 3i–l; mid-study), although glucose tolerance in both sexes did deteriorate over this period compared to normal chow (WT

male: ~500–700). After 16 weeks of HFD, there appeared to be a decline in glucose tolerance observed in adi-KO mice compared to control mice, which was not significant (genotype × time $p = 0.07$) in male adi-KO mice (Fig. 3n), but was significant (genotype $p = 0.018$, genotype × time $p = 0.029$) in female adi-KO mice (Fig. 3o). However, because the glucose load delivered for the oGTTs in these experiments was calculated based on total body mass, the larger size of the adi-KO mice resulted in them receiving a larger dose of glucose, confounding the kinetics of glucose disposal in these experiments. Therefore, we also performed oGTT experiments dosed according to LM, which demonstrated no significant difference in glucose tolerance between female WT and adi-KO mice (Supplementary Fig. 3A, B). These data confirmed that the greater dose of glucose in the initial GTTs was responsible for the observed differences in glucose tolerance. Finally, insulin tolerance tests (ITT) were performed after 12 weeks of HFD (Supplementary Fig. 3C to Supplementary Fig. 3F), which demonstrated no significant differences between any of the groups.

**Whole body energy expenditure suggests an altered metabolic flexibility in Trim28 adi-KO mice.** To determine the effect of Trim28 ablation on whole body substrate metabolism, we measured energy expenditure (EE) and respiration, using the Comprehensive Laboratory Animal Monitoring System (CLAMS). Characteristic changes in respiratory exchange ratio (RER; Fig. 4a, b, i, j), $VO_2$ and $VCO_2$ (Supplementary Fig. 4A to Supplementary

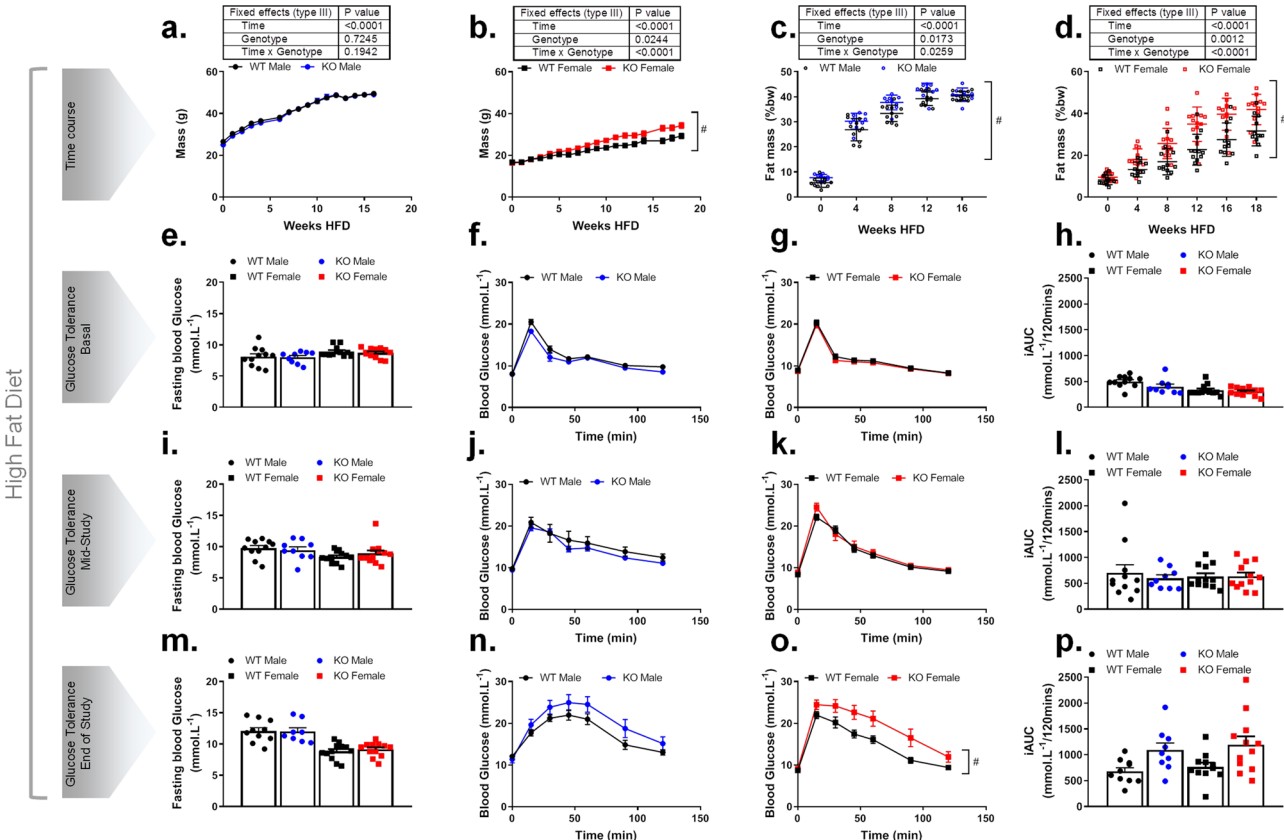

**Fig. 3 Trim28 adi-KO increases adiposity in mice fed a high-fat diet.** Trim28 wild type (WT) and adipose-specific Trim28 KO (KO) male and female C57BL/6 J mice were maintained on a high-fat diet (HFD) for 16 weeks from 6 weeks of age. Whole body adiposity measures including body mass and fat mass presented as percent body weight (BW) for (**a**, **c**) male mice and (**b**, **d**) female mice from 6 weeks of age at the commencement of HFD. Fasting blood glucose and oral glucose tolerance test in male and female mice with corresponding incremental area under the curve (iAUC) after 0 weeks of HFD (**e**–**h**; basal), 4 weeks of HFD (**i**–**l**; mid-study), and 16 weeks of HFD (**m**–**p**; end of study). All data are presented as mean ± SEM, female mice $n = 12$ WT and $n = 12$ KO, and male mice $n = 11$ WT and $n = 9$ KO. All data were analyzed using two-way ANOVA where *$p < 0.05$, or repeated measures mixed-effects model where #$p < 0.05$ for time × genotype interaction.

Fig. 4B and Supplementary Fig. 4G to Supplementary Fig. 4H, respectively) and EE (Fig. 4e, m) were observed across the light and dark periods; however, no significant differences were observed between male Trim28 WT and adi-KO mice on either diet.

However, for female mice there was a significant ($p < 0.05$) reduction in RER observed in Trim28 adi-KO mice during the light period compared with WT chow-fed mice (Fig. 4c, d), suggesting an increased preference to utilize fatty acids for metabolism at rest. No changes in $VO_2$ or $VCO_2$ were observed in female mice (Supplementary Fig. 4D to Supplementary Fig. 4E and Supplementary Fig. 4J to Supplementary Fig. 4K, respectively). Female HFD-fed adi-KO mice also demonstrated a reduction in RER compared with WT (not significant) during the light period, which was reversed during the dark period (Fig. 4k, l). This specific observation of a lower RER in female adi-KO mice during the day, but a higher RER during the night period, can be suggestive of a reduced capacity for substrate switching, often referred to as metabolic inflexibility.

In addition to these changes in RER, both chow- and HFD-fed female Trim28 adi-KO mice demonstrated significant changes in their pattern of EE, which was independent of both changes in LM (as analyzed by analysis of covariance (ANCOVA) $p = 0.018$ and $p = 0.04$, respectively, Fig. 4f, n) and total body mass ($p = 0.007$ and $p = 0.0008$ respectively, Supplementary Fig. 4F, L). No such differences were observed for male mice (Fig. 4e, m and Supplementary Fig. 4C, I), suggesting a sex-specific genotype

effect on energy metabolism. The most obvious differences in EE were observed in the female HFD-fed animals, where adi-KO animals had a higer EE in the setting of an equivalent amount of LM—particularly when the animals were smaller (Fig. 4n), indicating a subtle effect of adipose-specific Trim28 ablation on peripheral EE in female mice. This same trend was also apparent when we analyzed EE relative to total body weight (Supplementary Fig. 4L). No significant changes in total activity were observed between Trim28 WT and adi-KO mice of either sex or diet (Fig. 4g, h, o, p); although there was a trend for a reduced activity during the dark period for both male and female Trim28 adi-KO mice (Fig. 4g, o and Fig. 4h, p, respectively).

**Deletion of Trim28 in committed adipocytes alters lipid abundance.** To determine if the above changes in adiposity and metabolism occurred as a result of altered lipid abundance, we performed lipidomics (ESI-MS/MS) on plasma, liver, and gonadal WAT from WT and adi-KO chow-fed mice. We chose to focus on female mice due to their more robust phenotype (although we did analyze male data), and only those fed a chow diet in order to reduce spurious results that occur as a result of the large lipid insult induced by HFD feeding. Because the bulk of lipids in adipose tissue are triglycerides (TGs; Supplementary Fig. 5A, B), we focused on the regulation of these lipids. There was no change in the abundance of total TGs in WAT between genotypes (Supplementary Fig. 5C); however, we did observe consistent but

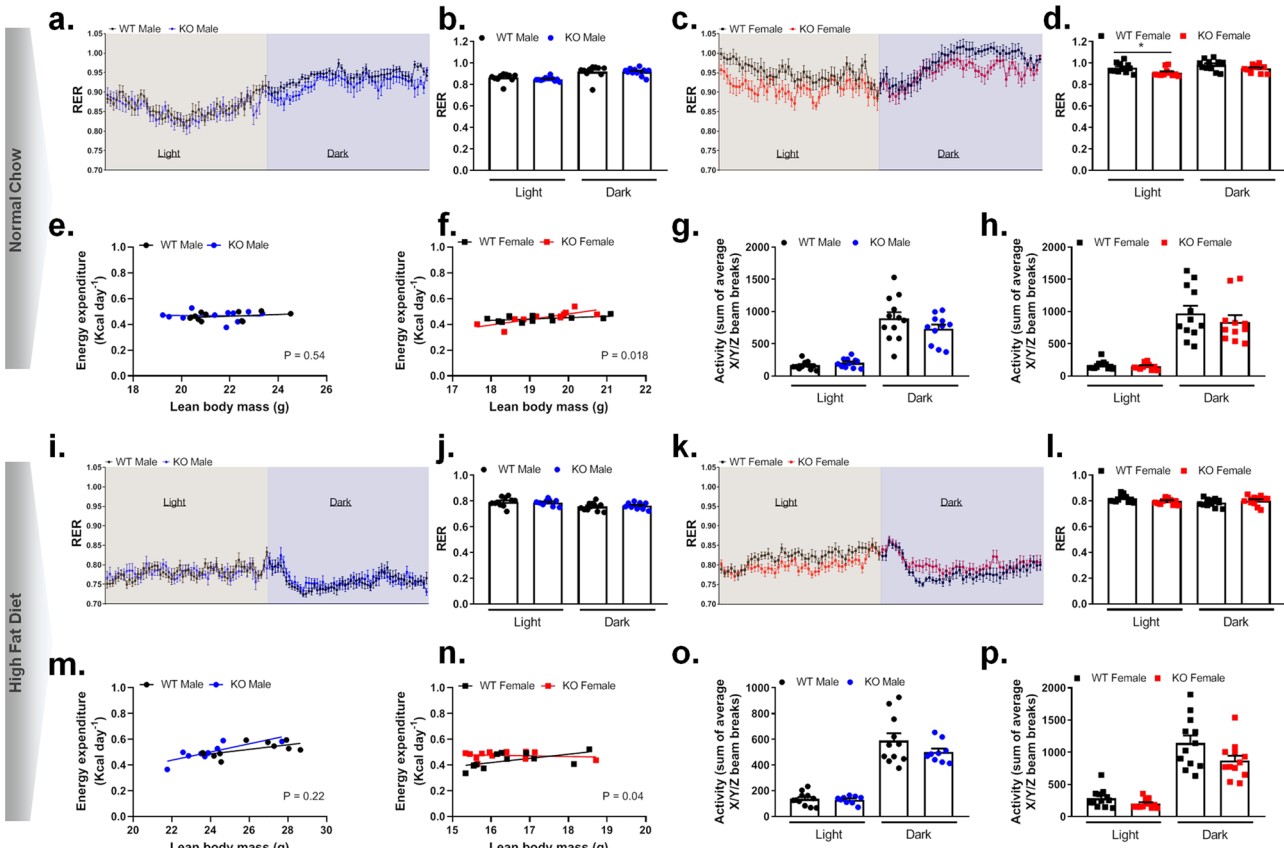

**Fig. 4 Whole body respirometry in male and female WT and adi-KO mice fed a chow or high-fat diet.** Trim28 wild type (WT) and adipose-specific Trim28 KO (KO) male and female mice were placed in metabolic cages (CLAMS) at 11 weeks of age (5 weeks post diet) for ~2 days. Respiratory exchange ratio (RER) over 24 h for chow-fed (**a**) male and (**c**) female mice, and average RER for light and dark periods for (**b**) male and (**d**) female mice. Energy expenditure assessed by ANCOVA (analysis of covariance) over 24 h for (**e**) chow-fed male and (**f**) female mice, and average activity, calculated as the sum of the average X/Y/Z beam breaks, for (**g**) male and (**h**) female chow-fed mice during light and dark periods. Respiratory exchange ratio (RER) over 24 h for (**i**) HFD-fed male and (**k**) female mice, and average RER for light and dark periods for (**j**) HFD-fed male and (**l**) female mice. Energy expenditure assessed by ANCOVA over 24 h for (**m**) HFD-fed male and (**n**) female mice, and average activity, calculated as the sum of the average X/Y/Z beam breaks, for (**o**) male and (**p**) female mice during light and dark periods. All data are represented as mean ± SEM, *$p < 0.05$ versus WT; $p < 0.05$ considered significant for ANCOVA analysis of energy expenditure. Chow-fed mice: female mice $n = 11$ WT and $n = 11$ KO, and male mice $n = 12$ WT and $n = 11$ KO. HFD-fed mice: female mice $n = 12$ WT and $n = 12$ KO, and male mice $n = 11$ WT and $n = 9$ KO. RER respiratory exchange ratio, (g) grams.

nonsignificant reductions in liver and plasma TGs in Trim28 adi-KO mice (Supplementary Fig. 5C, D). Although there were no changes in total TG abundance in WAT, a more detailed analysis demonstrated distinct changes in specific TG species from Trim28 adi-KO mice (Fig. 5a; WAT). Specifically, we observed a consistent alteration in the pattern of TG saturation (i.e., number of carbon double bonds), where within a given subclass (i.e., dots of the same color) there was an increase (shift to the right) in species containing saturated fatty acids (no double carbon bonds), and a decrease (shift to the left) in species containing poly-unsaturated fatty acids (several carbon double bonds). The most prominent of these effects was observed in the TG54:x subclass (red dots), where several saturated TGs [TG(54:0)–TG(54:2)] were significantly increased, whereas most unsaturated species [TG(54:7)] were significantly decreased. These findings suggest that Trim28 deletion in adipose tissue elicits consistent and specific effects on TG metabolism. With regards to liver and plasma, we did not observe the same changes in TG saturation, but they were loosely opposite that of WAT (Fig. 5b, c). Never-theless, there was an overall reduction (shift to the left) in the abundance of short chain TGs (TG48:χ to TG54:x) in both liver and plasma (for a complete list of TGs, see Supplementary Data Table 1). Given these are the most abundant TGs in mammalian tissues, including the liver (see Supplementary Fig. 6 for an

analysis by abundance), this represents a biologically significant change that is consistent with that observed when reducing hepatic steatosis[15]. There was no consistent trends in other lipid species, therefore, we have not discussed these further (see Sup-plementary Data Table 1 for details). Alterations to the TG composition observed in WAT, liver, and plasma of female mice, was to a lesser extent recapitulated in chow-fed male mice (Supplementary Fig. 7A).

**Pathways related to TG metabolism are altered in female Trim28 adi-KO WAT.** To gain insight into the molecular mechanisms underlying the observed phenotypes in female Trim28 adi-KO mice, we next analyzed specific pathways involved in adipocyte TG metabolism and storage. A major pathway that regulates TG metabolism is lipolysis, and although we did not observe transcriptional changes in lipolytic genes in WAT (*Acaca*, acetyl-CoA carboxylase—ACC or *Lipe*, hormone-sensitive lipase—HSL; Supplementary Fig. 8A), we did detect significant changes in their protein regulation. Specifically, we observed significantly reduced phosphorylation of HSL at serine 563 (s563) in adi-KO mice (Fig. 6a, b), consistent with an inhibition of HSL-mediated lipolysis. Moreover, we demonstrated that Trim28 adi-KO mice had significantly reduced ACC (tACC)

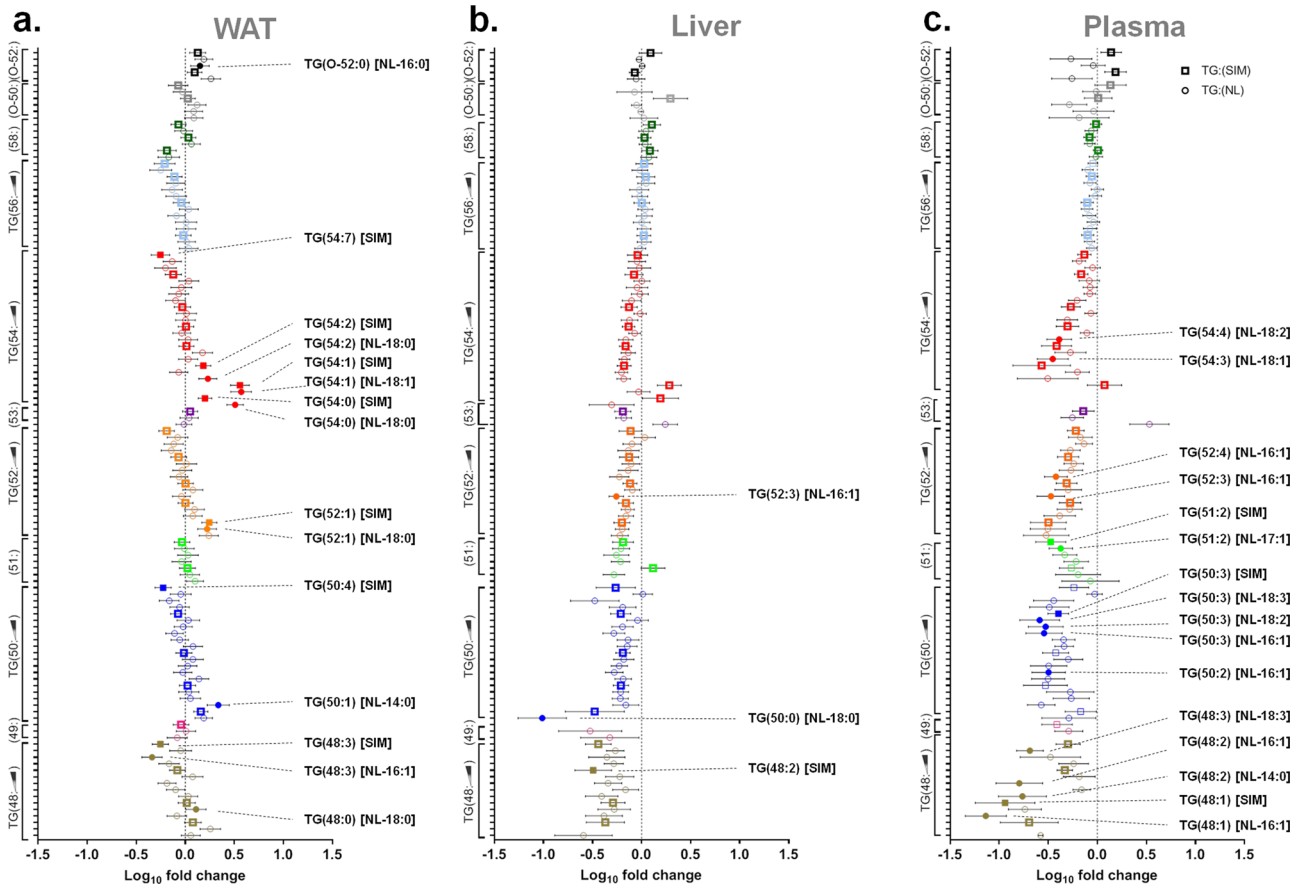

**Fig. 5 Trim28 adi-KO alters triglyceride composition in WAT.** Lipidomics in gonadal fat pads (WAT), liver, and plasma from female mice fed a chow diet for 24 weeks. Triglyceride (TG) species from (**a**) WAT, (**b**) liver, and (**c**) plasma. Triglyceride species were ordered into subclasses based on the combined FA carbon chain length (i.e., TG48 = 3× FAs containing 16 C each), then sorted vertically within those subclasses according to increasing carbon bond saturation (in the order indicated by the scale bar). For a complete list of triglycerides analyzed, see Supplementary Data Table 1. Squares represent data for single ion monitoring mode (SIM) of lipid species; circles represent species with annotation of a single FA side chain as determined by neutral loss analysis (NL). Data are expressed as $\log_{10}$ fold change from Trim28 WT mice as mean ± SEM, $n = 11$ WT and $n = 11$ KO. Individual comparisons were analyzed using Student's two-tailed unpaired $t$ tests. Closed symbols represent species that were statistically significant ($p < 0.05$) compared to WT. TG triacylglycerol, NL neutral loss, O-TG alkyl triacylglycerols.

protein abundance (Fig. 6a, c), a key protein involved in de novo lipogenesis and transport of FAs into the mitochondria for oxidation.

We next sought to determine whether the observed changes in lipolysis in Trim28 adi-KO mice were impacted by the effects of insulin. Akt phosphorylation is a key component of the insulin signaling pathway, and phosphorylation of the serine 473 site of Akt indicates activation of the insulin signaling pathway. We demonstrated that there were no significant changes in Akt phosphorylation in WAT (Fig. 6d, e) between WT and adi-KO mice, suggesting that these alterations in lipolytic signaling were independent of insulin. Consistent with this, we did not observe changes in other downstream components of the insulin signaling pathway, such as phosphorylation of GSK-3β at serine 9 (Fig. 6d, f). Finally, we investigated the effects of Trim28 deletion on the protein DJ-1, which has recently been implicated in the suppression of adipocyte β-oxidation[16,17]. We observed a significant ($p < 0.05$) increase in *Park7* (DJ-1) mRNA expression (Fig. 6g) and protein abundance in Trim28 adi-KO mice (Fig. 6h). Overall, these findings provide strong evidence for a role for Trim28 in regulating adipocyte lipid homeostasis through modulation of TG lipolysis and catabolism. We also observed a trend ($p = 0.1$) for a reduction (~35%) in total nonesterified (free) fatty acids (NEFA/FFA) in

plasma of adi-KO mice (Supplementary Fig. 8B), which appeared to be related to a specific subset of lower abundance FFAs (e.g., 14:0, 16:1, 20:2, and 20:3; Supplementary Fig. 8C).

To further validate these effects on lipolysis, we studied a 3T3-L1 cell line that was stably depleted for Trim28 using shRNAs (shT28), and compared them to a control stable line (shLuc). We demonstrated that Trim28 protein expression was significantly reduced in these cells (Supplementary Fig. 8D) and that no major differences were observed in the expression of early adipogenic proteins in these cells, or in their ability to lipid load once differentiated (Supplementary Fig. 8E). We subsequently treated these adipocytes with or without the adrenergic agonist isoproterenol (0–5 μM) for 6 h to activate lipolysis, and measured the release of glycerol. Minimal lipolysis was occurring basally in these cells, however, the Trim28-depleted cells exhibited significantly lower levels of glycerol release than control cells (Fig. 6i, 0 μM isoproterenol). Upon treatment with isoproterenol, we observed dose dependent increases in glycerol release in control cells, however, the Trim28-depleted adipocytes demonstrated a significant blunting of this response at both 0.5 and 5 μM of treatment ($p < 0.05$ for both; Fig. 6i). This was further supported by immunoblotting for pHSL-s563, which recapitulated the results observed on pHSL-s563 in vivo (Fig. 6j).

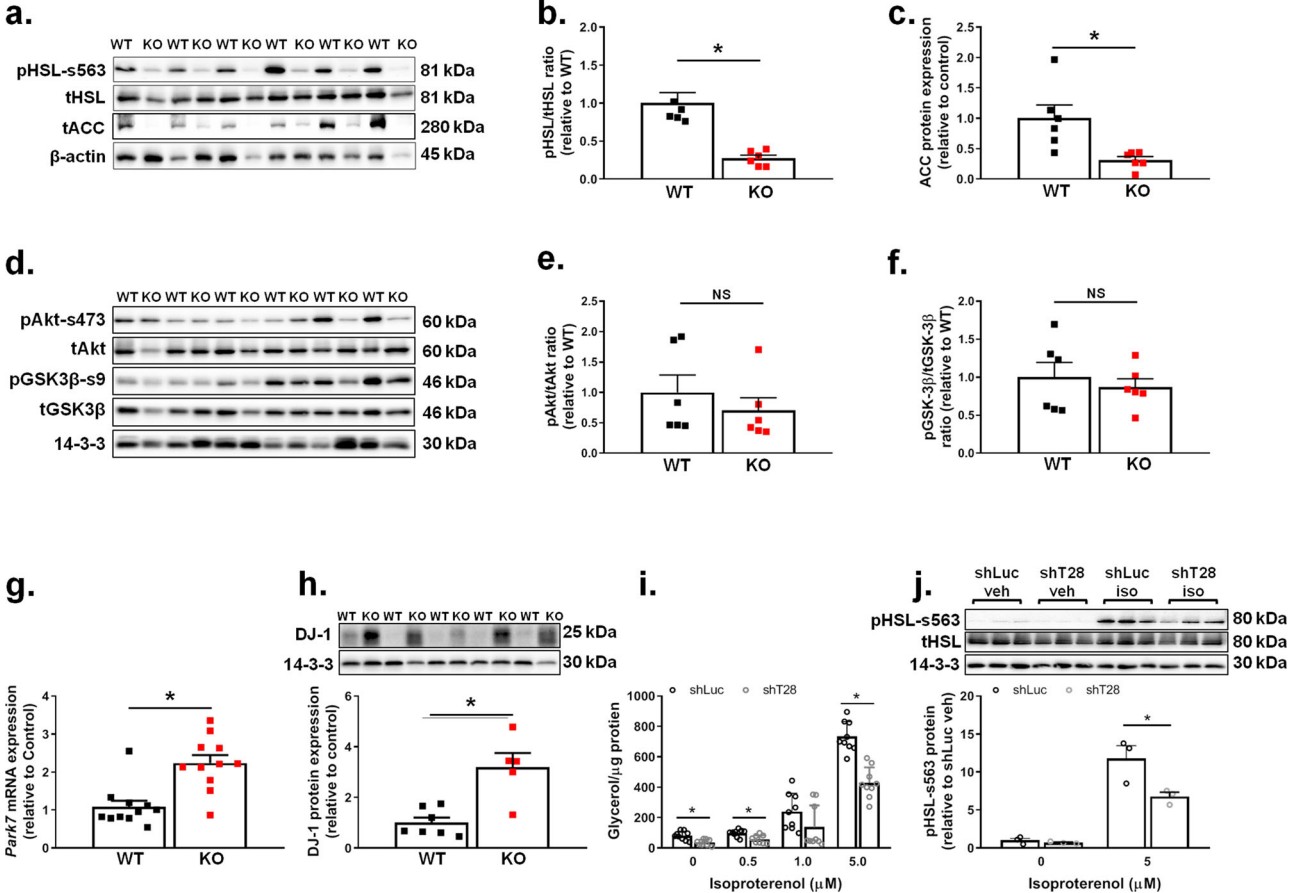

**Fig. 6 Trim28 adi-KO mice demonstrate altered pathways associated with adipocyte lipid storage and metabolism.** Protein analyses performed in gonadal fat pads from Trim28 wild type (WT) and adipose-specific Trim28 KO (KO) female mice fed a HFD. **a** Western blot from WAT of HFD-fed mice for serine 563 phosphorylation of HSL (pHSL-s563), total HSL (tHSL), total ACC (tACC), and beta-actin (β-actin). Densitometry quantification is shown for (**b**) HSL phosphorylation at serine 563 and (**c**) total ACC, ($n = 6$/group) *$p < 0.05$ versus WT mice. **d** Western blot from WAT of HFD-fed mice for serine 473 phosphorylation of Akt (pAkt-s473), total Akt (tAkt), serine 9 phosphorylation of GSK-3β (pGSK-3β-s9), total GSK-3β (tGSK-3β), and 14-3-3 (14-3-3). Densitometry quantification is shown for (**e**) Akt phosphorylation at serine 473 and (**f**) GSK-3β phosphorylation at serine 9, ($n = 6$/group for each). **g** mRNA expression of *Park7* (DJ-1, $n = 11$/group), and (**h**) DJ-1 protein expression as determined by western blot (above, $n = 6$), *$p < 0.05$ versus WT mice. Glycerol release and HSL serine 563 phosphorylation were measured in control (shLuc) and Trim28-depleted (shTrim28) 3T3-L1 cells in response to isoproterenol treatment (0–5 μM) as a measure of lipolysis. **i** Glycerol release normalized to mg protein in shLuc and shTrim28 3T3-L1 adipocytes at day 8 post differentiation stimulated with 0, 0.5, 1, and 5 μM isoproterenol ($n = 9$/group). **j** HSL serine 563 phosphorylation as determined by western blot (above) normalized to total HSL and relative to shLuc vehicle (veh: 0 μM) in shTrim28 3T3-L1 adipocytes. *$p < 0.05$ versus shLUC, $n = 3$/group in triplicate. All data are presented as mean ± SEM. All data were analyzed by ANOVA with post hoc testing (Fisher's LSD). ACC acetyl-CoA carboxylase, HSL hormone-sensitive lipase, DJ-1 protein and nucleotide deglycase DJ-1.

## Gene networks that regulate metabolism and lipid biology in adipocytes are altered in Trim28 adi-KO Mice.

We next performed gene expression analysis using qPCR on pathways that have previously been implicated with Trim28 function or lipid metabolism. This included gene sets previously described by Dalgaard and colleagues, as well as metabolic pathways, such as browning of WAT. Previous studies utilizing the global haploinsufficient Trim28 mice had proposed that the increased adiposity was due to alterations in genes from the nonclassical imprinted gene network 1 cluster, namely *Peg3* and *Nnat*[8]. Therefore, we assessed the expression of these genes in WAT of Trim28 WT and adi-KO female mice. These data demonstrated that there was no significant difference in the expression of either of these genes in any group (Fig. 7a, b), implying that IGN1 genes were unlikely to be driving the phenotype in our committed adipocyte model.

We also demonstrated that the majority of genes analyzed relating to lipid metabolism were also not altered in adi-KO WAT, with the exception of *Elovl3* (elongation of very long chain

fatty acids protein 3; Supplementary Fig. 9A). *Elovl3* was altered by up to 100-fold in various adi-KO adipose tissue depots (gonadal, inguinal, and brown; Supplementary Fig. 9B), which occurred irrespective of diet and sex (Fig. 7c, d). Given that *Elovl3* is often an indication of increased WAT browning, we investigated whether other genes involved in adipocyte browning were also increased in adi-KO mice. qPCR analysis indicated that there was no increase in the majority of genes related to browning of WAT (Fig. 7e), or increased activity in BAT (Fig. 7f). *Elovl3* is also a regulator of fatty acid elongation, in particular the synthesis of saturated and monounsaturated fatty acids, thus its increased expression in adi-KO WAT is consistent with the alterations observed in TG metabolism (Fig. 5). Thus, given that neither the imprinted genes nor browning appeared to be responsible for the altered phenotype in adi-KO mice, we performed RNA sequencing to gain more mechanistic insights.

RNA sequencing was performed on WAT from both Trim28 WT and adi-KO male and female mice fed a chow diet ($n = 6$/ group), which revealed many genes that were altered by Trim28

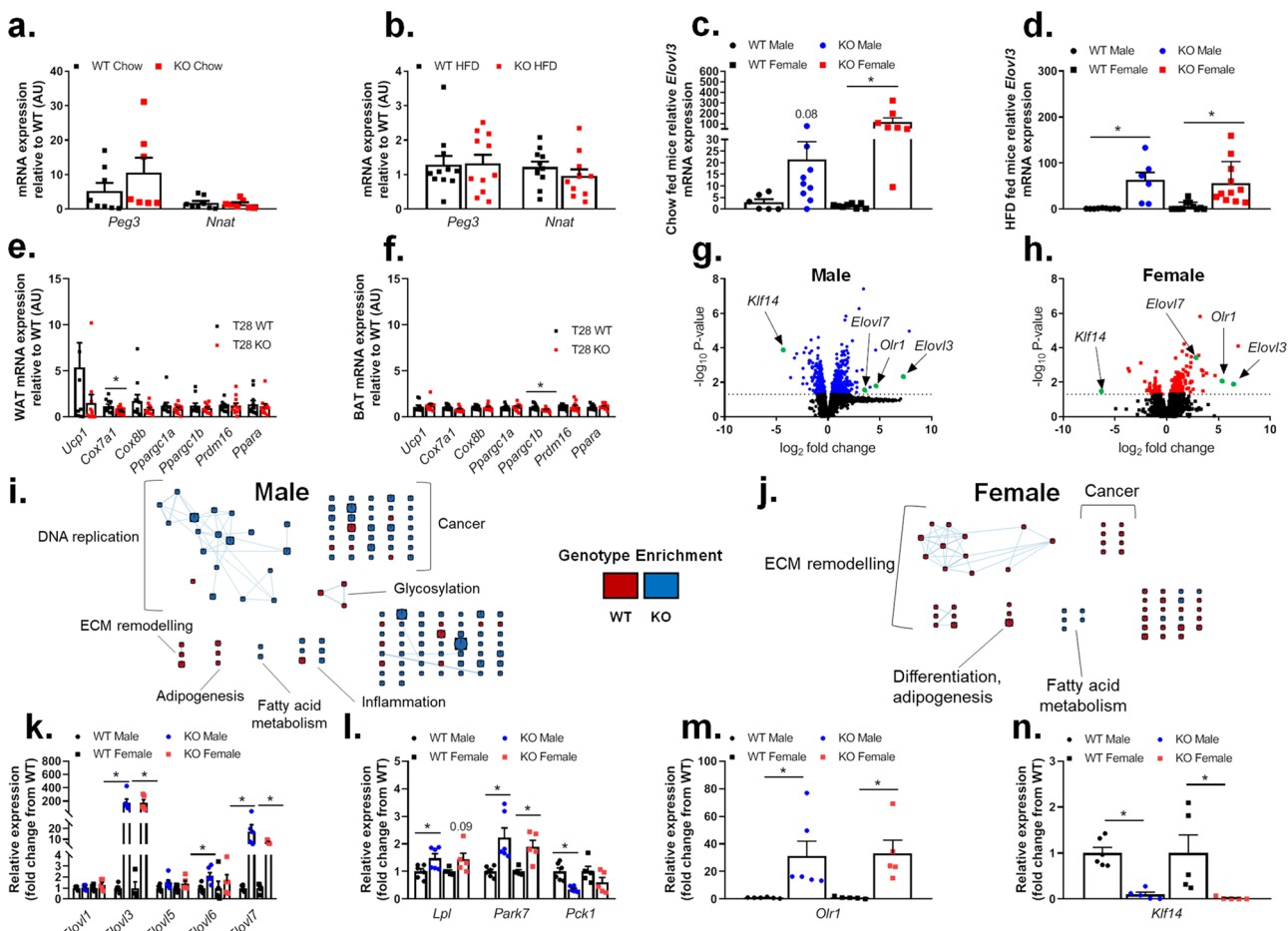

**Fig. 7 Trim28 adi-KO alters metabolic gene expression in WAT.** Gene expression analyses performed in gonadal fat pads from female Trim28 wild type (WT) and adipose-specific Trim28 KO (KO) mice fed either a chow or HFD. Imprinted gene network 1 (IGN1) genes *Peg3* and *Nnat* mRNA expression in (**a**) chow fed, $n = 8$ WT and $n = 7$ KO, and (**b**) HFD-fed mice as determined by qPCR, $n = 11$/group. *Elovl3* mRNA expression in male and female (**c**) chow-fed mice $n = 6$ M-WT and $n = 10$ M-KO, $n = 8$ F-WT, $n = 7$ F-KO, and (**d**) HFD-fed mice $n = 9$ M-WT and $n = 7$ M-KO, $n = 11$ F-WT, $n = 11$ F-KO. qPCR analysis of gene expression for (**e**) browning of WAT $n = 10$ WT and $n = 11$ KO, and (**f**) surrogate markers of BAT activity $n = 10$ WT and $n = 11$ KO. Volcano plots of differentially expressed genes regulated between chow-fed WT and Trim28 adi-KO mice as determined by RNAseq analysis in gonadal adipose tissue from (**g**) male mice and (**h**) female mice. Functional map of genes enriched in WAT from chow-fed Trim28 WT (red) and adi-KO (blue) mice in (**i**) male and (**j**) female mice. Enrichment results underwent MSigDB pathway analysis to determine enriched pathways. Gene sets were related by mutual overlap (edge width), where node size is proportional to the total number of genes in each set. Abundance of transcripts determined from RNAseq in WAT of WT and Trim28 adi-KO mice for (**k**) *Elovl* family members, and significantly regulated metabolic genes (**l**) *Lpl*, *Park7*, and *Pck1*, (**m**) *Olr1* and (**n**) *Klf14*, (RNAseq: males; $n = 6$, females; $n = 5$). All data are represented as mean ± SEM, *$p < 0.05$ versus WT mice. RNA-seq data were analyzed using ANOVA with FDR correction (Benjamini Hochberg). Specific gene comparisons were analyzed by ANOVA with post hoc testing (Fisher's LSD).

adi-KO, as indicated by volcano plots in Fig. 7g, h (also see; Supplementary Data Table 2). Pathway enrichment analysis of the differentially expressed genes demonstrated an enrichment for several pathways consistent with those identified by Dalgaard et al. (Cancer, DNA replication), but also in many pathways indicative of altered lipid metabolism and adiposity (adipogenesis, differentiation, fatty acid metabolism; Fig. 7i, j). Following a more specific analysis of these data, we observed significant alterations in *Park7* and *Elovl3* (Fig. 7k, l), validating our previous qPCR findings. We also observed alterations in other enzymes involved in fatty acid elongation and metabolism, such as several *Elovl* family members (*Elovl6* and *Elovl7*—Fig. 7k), as well as *Lpl* and *Pck1*, supporting the notion that Trim28 regulates genes in adipose tissue that alter adipocyte function and lipid abundance (Fig. 7l).

Two genes substantially influenced by Trim28 adi-KO were *Olr1* (oxidized low density lipoprotein receptor 1) and *Klf14*, both of which have been previously implicated in adipocyte

function[18–25]. We demonstrated a 30–40-fold increase in *Olr1* gene expression in both male and female adi-KO mice (Fig. 7m), and an almost complete ablation of *Klf14* expression in female and male adi-KO WAT (Fig. 7n). *Olr1* has been previously implicated in regulating lipid content in adipocytes, but hasn't been associated with sex-specific differences. The latter gene, *Klf14*, is of particular relevance to the current study, largely because it has been shown in several studies to be genetically and mechanistically linked to female-specific differences in adipocyte function and adiposity in both rodents and humans. Moreover, to our knowledge no bona fide transcriptional regulators of *Klf14* have been previously described, thus our data provides evidence that Trim28 may be a core component of the transcriptional machinery that regulates *Klf14* expression.

In summary, our data demonstrate that adipose-specific deletion of Trim28 resulted in increased adiposity and metabolic inflexibility in adipocytes, particularly in female mice, which impacted on adipocyte lipolysis signaling, but not on glucose

tolerance. We also demonstrated precise alterations in adipose tissue TG abundance, which coincided with alterations in pathways that regulate lipolysis, lipid storage, and oxidation, suggesting that Trim28 dampens the ability of committed adipocytes to appropriately adapt substrate utilization both in vitro and in vivo. Moreover, the sex-specific differences in adipocyte phenotypes are likely explained in part by the loss of expression of *Klf14* in adipose tissue, a gene that has been previously implicated in sex-specific alterations in adiposity and is a master regulator of gene networks that regulate adipocyte lipid metabolism[23,24].

## Discussion

Mechanisms that regulate healthy adipose tissue expansion and function are suggested to impart beneficial effects on whole body metabolism by reducing the deposition of toxic lipids in peripheral tissues[26]. In line with this, in the current study, we demonstrated that Trim28 deletion in adipose tissue resulted in increased adiposity, particularly in female mice, which was not associated with decrements in glucose homeostasis. These findings are consistent with previous studies demonstrating that global Trim28 haploinsufficient mice are obese and metabolically healthy[8,9]. Moreover, we demonstrated that this adipose-specific phenotype can be recapitulated in vitro, and that the in vivo phenotype is robust and does not present in a bistable fashion as does the global haploinsufficient model, challenging the previous paradigm that Trim28-induced obesity primarily stems from alterations in developmental cellular lineages.

Previous work demonstrated that lean and obese phenotypes observed in Trim28 haploinsufficient mice were determined via an imprinted gene network involving *Peg3* and *Nnat*, which were regulated to influence downstream mechanisms[8]. Although we did not specifically analyze the expression of these genes in progenitor cells, in our model of Trim28 KO in developed adipose tissue we did not observe any major changes in the expression of these genes in WAT; however, we did observe significant changes in key genes involved in adipose tissue development and lipid metabolism. To investigate the functional consequence of these changes, we utilized a lipidomics platform to demonstrate increases in saturated TGs and reductions in polyunsaturated TGs. To explain these differences in TGs, we demonstrated that the expression of the FA elongases *Elovl3*, *Elovl6*, and *Elovl7* were all elevated in adi-KO WAT, which have been shown to elongate saturated and monounsaturated fatty acids up to 24 carbons long[27,28]. Westerberg et al. demonstrated that *Elovl3* is an important regulator of endogenous synthesis of saturated very long chain fatty acids, consistent with the increase in *Elovl3* expression, and increased saturated and monounsaturated TGs observed in Trim28 adi-KO mice[27,28]. Moreover, *Elovl3* has also been shown to play a role in lipid recruitment, TG formation, and expansion of adipose tissue[27], reminiscent of the phenotype observed in Trim28 adi-KO mice. In addition, shorter chain TGs were reduced in the plasma and liver of Trim28 adi-KO mice from TG(48:*x*) through to TG(54:*x*) subclasses, with longer chain FA species being mostly unchanged. These alterations in lipid abundance are likely to improve the metabolic health of the liver. Indeed, we observed lower fasting blood glucose in adi-KO mice suggestive of a greater suppression of hepatic glucose production, which is consistent with the trends for increased insulin signaling in the liver.

In line with the effects of Trim28 adi-KO on TG abundance, we observed reductions in the phosphorylation of HSL at s563 in WAT of adi-KO mice and in Trim28-depleted adipocytes in culture, consistent with an inhibition of lipolysis. s563 on HSL is phosphorylated by protein kinase A, which increases the activity

of HSL and thus hydrolysis of stored TGs[29,30]. Thus, regulation of HSL activity in Trim28 adi-KO mice WAT suggests a consistent regulatory effect on lipolysis. This effect on lipolysis is unlikely to be a result of changes in adipose tissue insulin sensitivity, as we did not observe any difference in fasting Akt phosphorylation at serine 473 in WAT. In support of an insulin-independent effect, we also observed similar effects on lipolysis and s563 phosphorylation of HSL in cultured adipocytes depleted for Trim28, a model in which there was no exogenous insulin present during cell experimentation. These changes in HSL phosphorylation in adi-KO adipocytes coincided with reduced abundance of ACC, an enzyme that regulates de novo lipogenesis and the transport of FAs into the mitochondria for β-oxidation[31], and thus also plays a significant role in regulating cellular lipid abundance. These somewhat opposing results suggest an uncoupling of lipid storage and release in these cells, which is impacting the regulation of lipid abundance in Trim28 KO adipocytes. This is consistent with an inflexibility of adipocytes to coordinately regulate energy substrate pathways, and perhaps delay the cells ability to switch from storage to release in a timely manner. This metabolic inflexibility was also observed in vivo, and could be an important aspect of the KO phenotype. However, because this effect appeared to be insulin independent, we investigated other pathways in an attempt to explain this phenomenon.

We first investigated the protein DJ-1, a mitochondrial protein that has recently been shown inhibit adipocyte fatty acid oxidation[16,17]. We demonstrated upregulation of DJ-1 mRNA (*Park7*) and protein abundance in Trim28 adi-KO mice, and although the exact mechanisms by which DJ-1 participates in lipid metabolism are unknown, increases in DJ-1 in Trim28 adi-KO mice remain consistent with reduced lipolysis and increased lipid storage.

We also investigated other molecular mechanisms that might explain this alteration in FA metabolism and lipolysis, using whole-transcriptome analysis. These data identified several interesting findings, which were critical in our understanding of the role of Trim28 function in adipose tissue. Two genes that were of significant interest from this analysis were *Olr1* and *Klf14*. *Olr1*, also known as LOX1, has been identified as being associated with obesity, atherosclerosis and other metabolic conditions[19–21]. Indeed, studies from Olr1-KO mice have demonstrated inhibition of de novo lipogenesis via reductions in expression of fatty acid synthase (*Fasn*), stearoyl-CoA desaturase (*Scd1*), and ELOVL family member 6 (*Elovl6*)[21]. Moreover, other studies have implicated *Olr1* as a gene that regulates FM in several mammalian models[22,25], and that its' expression is regulated by PPARγ to control adipocyte cholesterol levels[20]. Thus, the substantial upregulation of *Olr1* in adipose tissue of Trim28 adi-KO mice, could explain several aspects of the observed phenotype, particularly those related to alterations in the abundance of specific lipid species.

With respect to the sex differences observed in the Trim28 adi-KO phenotype (more pronounced phenotype in females), it is highly likely that loss of *Klf14* expression in adi-KO mice is playing a major role. *Klf14* has been shown in two large-scale genetic studies to be a master regulator of adipocyte gene networks, in which regulatory variants at the *Klf14* locus influenced T2D risk via a female-specific effect on adipocyte size and body composition[23,24]. *Klf14* regulates a network of over 300 genes that influences adipocyte function, many of which are consistently altered in our transcriptomics dataset. Global *Klf14* loss of function studies in mice have demonstrated mixed findings with regards to its role in obesity and metabolism, with two separate studies demonstrating inconsistent phenotypes on adiposity and glucose tolerance[18,24]. However, given these phenotypes were characterized in global Klf14 deletion models, studies that

investigate the effect of adipose-specific Klf14 deletion may provide pertinent insights. Nevertheless, by way of validating KLF14 function in adipocytes, it was shown in human primary adipocyte cultures, that loss of KLF14 signaling resulted in an increase in adipocyte size and a reduction in lipogenesis in female models[24], consistent with our findings in Trim28 adi-KO mice. Whilst it is highly likely that much of our phenotype might be explained by the changes in Klf14 gene expression, how Trim28 is regulating Klf14 expression remains unclear, but indeed warrants further investigation.

Given that KLF14 is a Kruppel-like factor domain containing gene, and that one of the primary functions of Trim28 is to interact with Kruppel-associated box repression domains—it is interesting to speculate whether TRIM28 may interact with KLF14 in adipocytes either directly or indirectly via common co-regulatory complexes. To investigate this, we engaged publically available protein–protein interaction (PPI) datasets, and were able to perform in silico modeling analyses to demonstrate that KLF14 and TRIM28 share a common interaction with SIN3a, a critical protein in HDAC1 containing transcriptional repressor complexes (Supplementary Fig. 10). Other groups have also demonstrated that TRIM28 co-precipitates with complexes containing HDAC1 in HEK293 cells[32] and skeletal muscle myoblasts[33]. Thus, we speculate whether TRIM28, HDAC1, and KLF14 form an important transcriptional complex in adipocytes, which might become dysregulated if one of these factors, such as Trim28, is missing. While this is mostly speculative, importantly we demonstrate in the current study that Trim28 expression is necessary for the expression of Klf14, and provides a plausible explanation to the exacerbated female adiposity phenotype observed in the Trim28 adi-KO mice. In support of this association between Trim28 and Klf14, Dalgaard et al. also observed robust changes in Klf14 expression in their Trim28 haploinsufficient mice, however, this was not discussed as a potential gene to explain their phenotype[8].

Overall, in the context of previous work, our data demonstrate that adipocyte-specific Trim28 can modulate adipose tissue function, whereby reduced Trim28 expression promotes adiposity—potentially through modulating lipid switching and substrate preference—which does not lead to decrements in whole body glucose metabolism. Although food intake was not measured during this study, previous studies have suggested that Trim28 deletion in global and liver-specific models does not impact food intake[8,34], however, this is a limitation to the current study and should be measured in the future. Furthermore, because key adipocyte pathways, such as lipid catabolism and storage were impacted in an insulin-independent manner, we proposed that the loss of Klf14 expression impacts adipocyte metabolism in a sex-specific manner. This affected the composition of individual TG species in WAT, which subsequently resulted in a beneficial alteration in the lipid content in the liver and plasma and manifested as obesity without glucose intolerance.

## Methods

**Animals**. All animal experiments were approved by the Alfred Medical Research and Education Precinct (AMREP) Animal Ethics committee (E/1618/2016/B), and performed in accordance with the ethical guidelines set out by the National Health and Medical Research Council of Australia. Trim28 deletion was achieved using the Cre-Lox system. For conditional adipose ablation, Trim28 floxed mice (C57BL/6 J, Jackson Laboratories) were crossed with AdipoQ-Cre mice (C57BL/6 J background, gift from Prof Mark Febbraio) to generate both male and female cohorts of Trim28$^{fl/fl}$-AdipoQ-Cre$^{+/-}$ (Trim28 adi-KO) or Trim28$^{fl/fl}$-AdipoQ-Cre$^{-/-}$ (Trim28 WT; control). All mice were bred and sourced through the AMREP Precinct Animal Centre and randomly allocated to groups. After weaning at 4 weeks of age, wild type and adipose-specific Trim28 KO male and female mice were age and sex matched, then housed at 22° on a 12 h light/dark cycle with access to food (chow: Specialty feeds, Australia) and water ad libitum with cages changed weekly. Mice were allowed to acclimatize for 2 weeks prior to commencement of

the diet regiment and housed for the duration of the study to 24 weeks of age. For HFD cohorts (43% energy from fat, #SF04-001 Specialty Feeds), HFD was administered from 6 weeks of age for up to 18 weeks. Experiments across different diet interventions and gender cohorts were age matched.

**Insulin ELISA and insulin tolerance tests**. Intraperitoneal ITT (insulin @ 1 unit/kg LM) were performed at 18 weeks of age (12 weeks HFD) after a 5 h fast[35]. Briefly, blood glucose was measured at baseline before mice were injected IP with insulin, then subsequent measures of blood glucose were analyzed at 15, 30, 45, 60, 90 and 120 min post injection. Plasma insulin quantification was determined using Mouse Ultrasensitive Insulin ELISA kit (ALPCO, USA) performed according to the manufacturer's instructions using 25 μl of plasma.

**Glucose tolerance tests**. oGTT were performed at different time points (6, 10, and 18 weeks of age) at a dose of 2 g/kg calculated to total body weight. An additional oGTT was performed at 24 weeks of age in the HFD-fed female mice (18 weeks HFD) with a glucose dose of 2 g/kg calculated to LM as determined by EchoMRI. All oGTTs were performed after a 5 h fast[36,37]. Briefly, blood glucose was measured at baseline before mice were gavaged with glucose solution, then subsequent measures of blood glucose were analyzed at 15, 30, 45, 60, 90, and 120 min post gavage.

**EchoMRI**. Body composition analysis, including LM, FM, and free water, were measured using the EchoMRI 4in1 Body Composition Analyzer[37,38]. Live, conscious mice were place in the chamber and readouts of FM and LM were recorded at the designated periods throughout the study.

**Calorimetry**. CLAMS (Columbus Instruments, Columbus, Ohio) was performed at 11 weeks of age[37,38]. Mice were placed in CLAMS housing chambers for 72 h, and data recorded in the final 24 h of the experiment were used for analysis.

**Cell culture and in vitro lipolysis experiments**. 3T3-L1 cells were cultured in DMEM (GIBCO #11965-084) supplemented with 10% NBCS and maintained in 5% CO$_2$ at 37 °C. Cells were differentiated in DMEM supplemented with 10% FBS, 20 nM insulin, 50 nM GW1929, 0.5 mM IBMX, 1 μM dexamethasone for 48 h, and then cultured in DMEM supplemented with 10% FBS and 20 nM insulin for upto 10 days. 3T3-L1 adipocytes depleted for Trim28 (shTrim28) and control (luciferase-shLuc) were generated using commercially sourced lentiviral particles (Sigma MISSION particles) expressing shRNAs designed via the Broad Institutes RNAi Consortium database (TRCN0000302256 and SHC002V, respectively). Briefly, 50,000 undifferentiated 3T3-L1 cells were plated and exposed to lentiviruses (MOI = 20) in the presence of polybrene (10 μg/mL) for 24 h, before being changed in to normal growth medium (DMEM + 10% NBCS) overnight. The following day positive cells stably expressing shRNAs were selected in puromycin (4 μg/mL) for 4 days, before returning cells to normal growth media prior to differentiation. Lipolysis experiments were carried out on shLuc and shTrim28 3T3-L1 adipocytes at day 8 post differentiation. Cells were washed two times in PBS then incubated for 6 h in serum-free media supplemented with either vehicle, 0.5, 1, or 5 μM isoproterenol. Media was then collected for glycerol assay normalized to protein performed, as previously described[39].

**Cell microscopy**. shLuc and shTrim28 3T3-L1 cells were grown in 12 well plates to day 10 post differentiation to allow for lipid loading. Cells were then imaged using an inverted Olympus IX71.

**SDS–PAGE and immunoblot**. Tissues and cells were harvested and lysed in radio-immunoprecipitation assay buffer supplemented with protease and phosphatase inhibitors. Matched protein quantities were separated by SDS–PAGE and transferred to PVDF membranes. Membranes were blocked in 3% skim milk for 2 h and then incubated with primary antibody overnight at 4 °C: Trim28 (Cell Signalling), β-actin (Santa Cruz Biotech), total HSL (Cell Signaling Technologies), pHSL-s563 (Invitrogen), total ACC (Cell Signaling Technologies), DJ-1 (Cell Signaling Technologies), pAkt-S473 (Cell Signaling Technologies), total Akt (Cell Signaling Technologies), pGSK3β-s9 (Cell Signaling Technologies), total GSK-3β (Cell Signaling Technologies), pan 14-3-3 (Santa Cruz), C/EBPα (Cell Signaling Technologies), and PPARγ (Cell Signaling Technologies). After incubation with primary antibodies, membranes were probed with their respective HRP-conjugate secondary anti mouse or anti rabbit (Biorad) antibodies in 3% skim milk for 2 h at room temperature, then visualised with chemiluminescence (Pierce). Approximated molecular weights of proteins were determined from a co-resolved molecular weight standard (BioRad, #1610374). The Image Lab Program was used to perform densitometry analyses, and all quantification results were normalized to their respective loading control or total protein.

**Quantitative PCR**. RNA was isolated from tissues using RNAzol reagent and isopropanol precipitation. cDNA was generated from RNA using MMLV reverse transcriptase (Invitrogen) according to the manufacturer's instructions. qPCR was

performed on 10 ng of cDNA using the SYBR-green method on an ABI 7500, using primer sets as outlined in Supplementary Data Table 3. Quantification of a given gene was expressed by the relative mRNA level compared with control, which was calculated after normalization to the housekeeping gene Cyclophilin a (*Ppia*) using the delta-CT method. Primers were designed to span exon–exon junctions and were tested for specificity using BLAST (Basic Local Alignment Search Tool; National Centre for Biotechnology Information). Amplification of a single amplicon was estimated from melt curve analysis, ensuring only a single peak and an expected temperature dissociation profile were observed.

**Lipidomics and lipid abundance**. Lipidomics was analyzed in plasma, adipose, and liver from female Trim28 WT and adi-KO mice on a normal chow diet, as described previously in detail[40]. Briefly, lipids were extracted from tissues/plasma using a modified Folsch extraction method, dried, and dissolved before application to ESI-MS/MS analysis. Quantification of lipids from MS analysis was performed using Mass Hunter Software (Agilent). FFA abundance was analyzed from mouse plasma using the WAKO NEFA kit according to the manufacturer's instructions, or by ESI-MS/MS following lipid extraction in glass vials.

**RNA-sequencing analysis**. RNA was isolated from gonadal WAT using RNAzol reagent and purified using RNA isolation columns according to the manufacturer's instructions (Zymo Research). RNA integrity was evaluated using the Agilent Tape Station 2200 according to the manufacturer's instructions (Agilent). RNA libraries were prepared using Kapa Stranded RNA-seq kits on samples with a RIN > 0.8 according to the manufacturer's instructions (Roche). Library quantities were determined using QUBIT (ThermoFisher), and equal amounts of all libraries were pooled and run across two lanes, using an Illumina HiSeq 2500 Sequencing System. A total of 24 de-multiplexed raw FASTQ files, containing ~50 bp single end reads were generated, with one low read depth (<1 million) sample removed. The remaining 23 samples achieved a read depth of ~22.29–52.22 million reads. These details and the raw data can be found at ArrayExpress with the accession E-MTAB-9809. The qualities of the raw sequence reads were assessed using FastQC version v0.11.7 (ref. [41]). Based on the quality reports, the read length varied between 35–50 bp across all the samples. Reads <45 bp, ranging from 0.3 to 8.0 % across all samples, were removed prior to alignment. The remaining reads were aligned to mouse (mm10) reference genome, downloaded, and indexed from the UCSC Genome Browser, using STAR aligner version 2.7.1a (ref. [42]). The resulting BAM files containing the aligned reads were provided to feature Counts version 1.6.2 (ref. [43]) to obtain gene-level read counts using the reference annotation file (GTF format downloaded from the UCSC browser). Lowly expressed genes, defined as having less than an average (average of all WT and KO samples) of 30 counts per million (CPM) reads, were filtered out. In total, 12,940 genes were retained for differential gene analysis, which was performed using DESeq2 and iDEP.90 (refs. [44,45]). Enrichment analysis and gene ontology was performed using the Database for Annotation, Visualization and Integrated Discovery (DAVID v.6.8) hosted by the National Institute of Allergy and Infectious Diseases (NIAID), NIH, USA[46]. Cluster analysis from RNA-sequencing data were derived from gene sets significantly altered between WT and KO animals. These datasets were analyzed using GSEA (v3.0)[47,48], and enrichment and cluster analysis were mapped to a network of the curated MSigDB C5 gene set collection[47,49].

**In silico modeling of protein–protein interaction datasets**. PPI datasets were downloaded through the String portal, which procures data from publically available third party databases. Using human PPI datasets, an independent search was performed for both KLF14 and TRIM28, with stringency set to allow two shells of interaction with no >10 and 5 interactors, respectively. This generated a list of 1501 bait (node 1) and prey (node 2) interactions for KLF14, and 2981 for TRIM28 (Supplementary Data Table 4). Each set was exported and filtered so that each bait and prey existed only once in each list, enabling the identification of common proteins (interactors) between the lists. Common proteins were identified by overlaying node 1 and node 2 in a crossover fashion between the KLF14 and TRIM28 datasets. This analysis identified seven proteins (CREBBP, EP300, SIN3A, HDAC1, PHF12, NCOR1, and TP53) that were consistently identified across all datasets regardless of how they were overlayed, and two proteins (SUDS3 and MXD1) which were consistent in only two and one datasets, respectively. SIN3A was the only protein to be found to be a direct interactor with both KLF14 and TRIM28. These proteins were manually annotated into the String network to generate the integrated figure shown in Supplementary Fig. 10.

**Statistical analyses**. All data were expressed as mean ± standard error of the mean (SEM). All statistical analyses in animal studies were analyzed by repeated measures two-way ANOVA/repeated measures mixed-effects model. Lipidomics, tissue analysis and cell-based experiments were analyzed by ANOVA with post hoc testing (Fishers LSD) where appropriate, or paired Student's *t* test unless otherwise stated. Analyses were performed using PRISM8 software and a *p* value of $p < 0.05$ was considered statistically significant.

**Data inclusion and exclusion criteria**. For animal experiments, phenotyping data points were excluded using predetermined criteria if the animal was unwell at the

time of analysis, there were identified technical issues (such as failed gas sensors in CLAMS), values were biological implausible (such as RER = 2.0) or data points were identified as outliers using Tukey's Outlier Detection Method (1.5IQR < Q1 or 1.5IQR > Q3). If repeated data points from the same mouse yielded data points that were outliers as per Tukey's rule, the entire animal was excluded from that given analysis (i.e., during glucose/ITT, indicating inappropriate injection or gavage). For in vivo and in vitro tissue and molecular analysis, data points were only excluded if there was a technical failure (i.e., poor RNA quality, failed amplification in qPCR, and failed injection in mass spectrometer), or the value was biological improbable. This was performed in a blinded fashion (i.e., on entire datasets before genotypes were known). RNA-seq data inclusions and exclusions are described above.

**Reporting summary**. Further information on research design is available in the Nature Research Reporting Summary linked to this article.

## Data availability
All data are available within the provided Supplementary Data Tables. Raw RNA-seq data can be found at ArrayExpress (accession E-MTAB-9809). Gene set enrichment analysis and gene ontology was performed using the Database for Annotation, Visualization and Integrated Discovery (DAVID v.6.8; https://david.ncifcrf.gov/). Network analysis of protein interaction data was performed using String (https://string-db.org/). Source data are provided with this paper.

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

## Acknowledgements

We acknowledge funding support from the Victorian State Government OIS program to Baker Heart and Diabetes Institute. These studies were supported by funding from the National Health and Medical Research Council (NHMRC) of Australia (to BGD APP1128060). B.G.D. and A.C.C. were supported by National Heart Foundation of Australia, Future Leader Fellowships (101789 and 100067, respectively). We thank all members of the MMA, LMCD, and Metabolomics laboratories at BHDI for their ongoing contributions. We also thank Prof. Mark Febbraio (Baker Institute) for providing us with AdipoQ-Cre mice, and are grateful for the assistance of Angela Cheng (UCLA) for help with RNA library preparation and sequence analysis.

## Author contributions

B.G.D. designed and conceived the study. S.T.B. and B.G.D. wrote the manuscript. B.G.D., S.T.B., E.J.K., A.T., S.C.M., C.Y., and Y.L. performed all experiments. N.M., S.T.B., Y.L., and P.J.M. performed lipidomics. E.J.T. and T.Q.A.V. prepared RNA and ran RNA sequencing, and A.P.N. and M.I. performed RNA-sequencing genome alignments, quality control analysis, and gene counts. D.C.H. and A.C.C. provided reagents, experimental advice, and access to resources. All authors read and edited the manuscript.

## Competing interests

The authors declare no competing interests.
