## [Peer Review File · Nature Communications]

Reviewers' comments:

Reviewer #1 (Remarks to the Author):

The paper describes the role of Trim28 in obesity and metabolic health. The authors demonstrate that adipocyte specific loss rather than a developmental effect leads to increased adiposity which seems to be sex specific independent of alterations in metabolic control. Interestingly this quite well phenocopies the human situation. The paper is very well done and the experiments are really performed at a high standard. I would ask that some of the conclusions regarding the influence of food intake and energy expenditure on obesity are revisited and rewritten. Given the small effect size the methodology employed here is not sufficiently sensitive to exclude either EE or food intake as a reason for the obesogenic phenotype and therefore some of the statements should be changed.

Reviewer #2 (Remarks to the Author):

In this study, Bond et al. investigated the role of Trim28 within the adipocyte population in mouse. Using a mouse model of Trim28 KO specifically in adipocytes, the authors showed a mild increase in fat mass, quite specific to the female gender. While the authors associated this phenotype with alterations of triglyceride metabolism and impaired lipolysis, so far, the mechanism remained vastly unexplored. While of potential interest given that Trim28 action on adiposity is thought to be related to imprinted gene networks, the study in its current form appears premature and lacks mechanistic insights to explain the observed phenotype. Some data are not appropriately analysed which cast doubts regarding the conclusions drawn by the authors.

Major comments:

1. The authors described Adiponectin-Cre Trim28^{fl/fl} (KO) phenotype without validating first their model, a validation that comes appears later in the manuscript and only in female mice. The authors should present the global gene expression profile of Trim28 in different tissues (to validate AT specificity) and adipocytes in both male and female. In addition, Figure 4A strongly suggests that other adipose cells express Trim28 and therefore might contribute to Trim28 previously described effect (Dalgaard et al., Cell, 2016). The authors should identify which other cell population in AT express Trim28.
2. The study initial observation is that Adiponectin-Cre Trim28^{fl/fl} (KO) mice are fatter than none Cre expressing animals (WT). The authors assume that their phenotype is developed at the adult age because adiponectin is not express in adipose progenitor cells and therefore did not influence development. However, this assumption appears quite uncertain given the recent publications showing expression of adiponectin in foetal adipose progenitors in E16.5 embryos (PMID: 26243869

and 28123942) and therefore could partially recapitulate the phenotype previously observed by Dalgaard et al. It would be important to investigate Trim28 expression during adipogenesis in KO progenitors. The body weight data at different times (notably the ones when fat mass has been evaluated) should be expressed as single mouse values to clarify the distribution of the body weight values within each genotype.

3. The authors described lower fasting glucose in female KO but did not further characterise this phenotype. How are the insulin levels in these mice? Did the authors measure some neoglucogenesis markers in the liver of the KO mice? How is the glucose level in feeding state in Trim28-KO mice?

4. The authors show that females, and potentially male, Trim28-KO mice have an increase fat mass, both in chow and HFD-fed condition, without affecting lean mass. It is surprising that, while the fat mass is expressed as percentage of body weight, the lean mass is expressed in grams. The manuscript would gain in clarity if the authors could be consistent in the way they express their body repartition data.

5. The statistic section is unclear, the used tests should be mentioned in the figure legends and the results of the 2-way ANOVA mentioned (rather than the result of a potential posthoc test). It is informative, notably regarding the differences in BW or fat mass to know if the differences observed are the result of a genotype effect only or an interaction between genotype and time/diet.

6. The energy balance phenotyping has been poorly designed and analysed. The food intake has not been evaluated and the energy expenditure data has not been analysed properly. The authors should perform a more specific analysis of the energy balance in their model in order to clearly define why these animals are fatter. May we suggest the following publications as guidelines to assess EE (PMID: 22205519) and food intake (PMID: 22751256).

7. The RER data is not interpreted. First RER values variate between 0.7 (pure fatty acid oxidation) and 1 (pure glucose oxidation), the graphs of RER should be plotted from a higher value than 0 in order to appreciate better the effect size. It is worth mentioning that this mouse model may have a metabolic inflexibility phenotype, potentially as a result of an impaired lipolysis (as suggested by the p-HSL blot in Fig. 6 or the lipidomic data). However, the reduced RER may suggest the opposite. Can the authors provide fed and fasted FFA and TG levels to further explore this aspect? Additionally, markers of lipid uptake by adipose tissue should be also presented. Recent publications have shown the importance of adipose tissue in the regulation of metabolic flexibility (PMID: 29789270 and PMID: 30134163).

8. The authors claim that Trim28 deletion increases BW without affecting glucose metabolism. The GTT provided in Fig. 2K-L however suggests the opposite. If the authors want to make such strong statement, they should assess glucose homeostasis and insulin sensitivity using state of the art techniques such as hyperinsulinemic-euglycemic clamp.

9. Why the adipose tissue may be different is unclear at this stage. Depots mass, histology of the tissues, notably assessing adipocyte size, may help as well as RNAseq to define the pathways altered in Trim28 KO may guide the authors toward a potential mechanism that could explain their phenotype.

10. The lipidomic data show that Trim28-KO mice have alteration in chain length and desaturation of their TG that are somehow mirrored in the serum. Can the authors provide gene expression data regarding more elongases than Elovl3 (which is a browning marker) and of the fatty acid desaturases (SCDs, FADS).

11. It is quite surprising to observe such an increase in Elovl3 in ScWAT, this data can support a browning phenotype of the WAT and a potential increased SNS tone. Given that adiponectin is also expressed in BAT and therefore Trim28 must be deleted in BAT adipocytes, it may have a functional impact. Can the authors provide gene expression analyses of the BAT, ScWAT and gWAT of the thermogenic markers (e.g. UCP1, Dio2, Elovl3, etc.)?

The impaired lipolysis is so far the clearest result of the study and should be supported by a proper lipolysis experiment in AT from female KO mice. Are the mice resistant to adrenergic agonist? Is this phenotype also found in KO male? As for comment 7, fed and fasted FFA and TG levels will help to confirm the potential lipolytic phenotype.

12. The authors claimed the obese KO mice to be healthier. Are the mice protected from liver steatosis ?

Minor comments:

1. Acaca is not a “gene involved in adipose tissue lipolysis”, it is the first step of de novo lipogenesis and ACC is actually a target of AMPK. AMPK phosphorylates ACC in condition of energy deprivation to reduce DNL and increase FAO. Additionally, there is not a reduction in ACC phosphorylation but a general reduction in ACC protein. The authors should amend their manuscript to avoid any confusion regarding this pathway.

2. The authors should present GTT results from male and female in two different graph to facilitate the interpretation.

3. The n number is wrong in the legend of figure 6. The n numbers are actually different from one panel to another.

Bond et al., NCOMMS-19-17559: Response to Reviewers

Reviewer #1

The paper describes the role of Trim28 in obesity and metabolic health. The authors demonstrate that adipocyte specific loss rather than a developmental effect leads to increased adiposity which seems to be sex specific independent of alterations in metabolic control. Interestingly this quite well phenocopies the human situation. The paper is very well done and the experiments are really performed at a high standard. I would ask that some of the conclusions regarding the influence of food intake and energy expenditure on obesity are revisited and rewritten. Given the small effect size the methodology employed here is not sufficiently sensitive to exclude either EE or food intake as a reason for the obesogenic phenotype and therefore some of the statements should be changed.

We thank reviewer #1 for their positive remarks, and for their constructive comments regarding energy expenditure and food intake. We have now addressed these concerns by both reanalysing the data based on both reviewer comments, and by rewording some of our sections relating to the interpretation of this data. Please see Figure 4 and Supplementary Figure S4; and lines 224-252 in the main text.

Reviewer #2

In this study, Bond et al. investigated the role of Trim28 within the adipocyte population in mouse. Using a mouse model of Trim28 KO specifically in adipocytes, the authors showed a mild increase in fat mass, quite specific to the female gender. While the authors associated this phenotype with alterations of triglyceride metabolism and impaired lipolysis, so far, the mechanism remained vastly unexplored. While of potential interest given that Trim28 action on adiposity is thought to be related to imprinted gene networks, the study in its current form appears premature and lacks mechanistic insights to explain the observed phenotype. Some data are not appropriately analysed which cast doubts regarding the conclusions drawn by the authors.

We thank reviewer #2 for their detailed assessment of our manuscript, which provides many insightful suggestions. We have endeavoured to address all of the major concerns, which we believe has considerably strengthened the manuscript and conclusions in our paper.

Major comments:

1. The authors described Adiponectin-Cre Trim28^{fl/fl} (KO) phenotype without validating first their model, a validation that comes appears later in the manuscript and only in female mice. The authors should present the global gene expression profile of Trim28 in different tissues (to validate AT specificity) and adipocytes in both male and female.

We apologise for this oversight, and now include validation of the KO model in Figure 1. We also include new data demonstrating that Trim28 is reduced in cultured adipocytes as they differentiate into fully lipid loaded adipocytes. This provides further evidence that Trim28 is robustly expressed in progenitors and immature adipocytes, but is consistent with our hypothesis that loss of Trim28 is associated with lipid loading in adipocytes.

In addition, Figure 4A strongly suggests that other adipose cells express Trim28 and therefore might contribute to Trim28 previously described effect (Dalgaard et al., Cell, 2016). The authors should identify which other cell population in AT express Trim28.

We agree with the reviewer that only cells expressing adiponectin will have deletion of Trim28. As discussed in more detail below, adiponectin and thus cre-recombinase in this model will therefore be expressed in all cells committed as being adipocytes. This will include newly committed adipocytes which have yet to fully develop and store lipid (i.e. pre-adipocytes, but not precursors). Thus, in theory a fat depot *in vivo* would almost certainly contain a whole spectrum of adipocytes ranging from newly committed through to fully developed and full of lipid, all of which would be deleted for Trim28. Adipose tissue also contains several other cell types such as precursors, immune, structural and vascular cells (commonly referred to as the stromal vascular fraction), which of course would not be deleted for Trim28. Thus, we believe this latter pool of cells would account for the residual expression of Trim28 observed in both our gene expression and protein expression analyses.

2. The study initial observation is that Adiponectin-Cre Trim28^{fl/fl} (KO) mice are fatter than non Cre expressing animals (WT). The authors assume that their phenotype is developed at the adult age because adiponectin is not expressed in adipose progenitor cells and therefore did not influence development. However, this assumption appears quite uncertain given the recent publications showing expression of adiponectin in foetal adipose progenitors in E16.5 embryos (PMID: 26243869 and 28123942) and therefore could partially recapitulate the phenotype previously observed by Dalgaard et al. It would be important to investigate Trim28 expression during adipogenesis in KO progenitors.

Thank you for this comment, and we apologise for the confusion relating to this interpretation of the model. We did not mean to suggest that adiponectin is only expressed in adult mice, because as discussed above in detail, we are aware that adiponectin is expressed in all cell types that are committed to being adipocytes, whether fully mature or not. The idea that we were trying to convey was that the obesity phenotype observed by Dalgaard et al, is almost completely recapitulated by deleting Trim28 in committed adipocytes (where adiponectin is expressed) - which are not usually considered developmental cell types - and not from a deletion that occurred in more immature stem-like cells. Indeed, this is supported by the evidence that adiponectin expression is not present in early development in utero, and is only present in the final trimester (your reference: PMID: 26243869.)

Dalgaard et al states in their paper that *“Importantly, conditional homozygous deletion of Trim28 in muscle (Mck-Cre), adipose (Adipoq-Cre), liver (Alb-Cre), and the satiety regulating POMC- (POMC-Cre), and AgRP-neurons (AgRP-Cre) have revealed no obvious effect on adiposity. These data indicate that TRIM28 is largely dispensable in fully differentiated adult tissues and support a role, consistent with much literature, in transcriptional programming in development.....”* Our data clearly demonstrate that Trim28 is not dispensable in fully differentiated tissues (a committed adipocyte is considered differentiated). Whilst we will concede that committed adipocytes are not necessarily fully differentiated adult tissues, they are also not developmental cell types either.

Nevertheless, in light of these considerations we have changed the wording in our manuscript when referring to the specific cell types that express adiponectin and thus cre, and have loosened the terminology around claims that we deleted Trim28 in “fully developed” adipose tissue.

Regarding the expression of Trim28 during adipogenesis, this is now shown in Figure 1.

The body weight data at different times (notably the ones when fat mass has been evaluated) should be expressed as single mouse values to clarify the distribution of the body weight values within each genotype.

We have now replotted the data to include individual data points, to better demonstrate the distribution. See Figures 2C&D (normal chow) and 3C&D (high fat diet).

3. The authors described lower fasting glucose in female KO but did not further characterise this phenotype. How are the insulin levels in these mice? Did the authors measure some neoglucogenesis markers in the liver of the KO mice? How is the glucose level in feeding state in Trim28-KO mice?

We have now further characterised the glucose phenotype. We analysed fasting plasma insulin levels and did not observe any difference between groups (Supp Fig S1I), indicating that insulin secretion/clearance is not driving this difference. This suggests that the small but detectable improvements in fasting blood glucose are likely due to improved insulin sensitivity. Indeed, in support of this hypothesis we demonstrated that phosphorylation of Akt at Ser473 (a site associated with increased activity) in the liver was higher in the majority of adi-KO mice (see Supp Fig S1J), consistent with an increased insulin sensitivity in chow fed mice.

4. The authors show that females, and potentially male, Trim28-KO mice have an increase fat mass, both in chow and HFD-fed condition, without affecting lean mass. It is surprising that, while the fat mass is expressed as percentage of body weight, the lean mass is expressed in grams. The manuscript would gain in clarity if the authors could be consistent in the way they express their body repartition data.

We can understand the confusion from the reviewer regarding the presentation of this data, however this is in fact deliberate. Because the increase in body weights in KO mice is almost exclusively due to increases in fat mass, it is appropriate to express fat mass as a proportion of body mass, because the other components of body mass (i.e. lean mass) are not changing. Moreover, presenting the data adjusted for total body mass, eliminates much of the natural variation that exists between mice of a given genotype, thus allowing us to visualise the true change in fat mass more accurately. However, if we do this for lean mass it appears as if lean mass is decreasing as the mice age (See Supp Fig S2F) - which is not true, because the body mass is increasing as a result of increased fat mass. Therefore, we have chosen to present this data in the main figures as fat mass as a percentage of body weight. However, in the interests of transparency we have now also included all other measures of this data, including absolute mass, in Supplementary Figures S1 & S2.

5. The statistic section is unclear, the used tests should be mentioned in the figure legends and the results of the 2-way ANOVA mentioned (rather than the result of a potential posthoc test). It is informative, notably regarding the differences in BW or fat mass to know if the differences observed are the result of a genotype effect only or an interaction between genotype and time/diet.

We have now performed these analyses (repeated measures mixed effects model) and included these outputs in the text and in the figures where appropriate. See Figures 2 & 3 and Supplementary Figures S1 & S2.

6. The energy balance phenotyping has been poorly designed and analysed. The food intake has not been evaluated and the energy expenditure data has not been analysed properly. The authors should perform a more specific analysis of the energy balance in their model in order to clearly

define why these animals are fatter. May we suggest the following publications as guidelines to assess EE (PMID: 22205519) and food intake (PMID: 22751256).

Thank you for your assessment and for providing guidance relating to our energy balance studies. We respectfully disagree that they have been poorly designed and analysed, but our group and we suspect the field in general, are always looking for more consistent ways to analyse and present these data.

We take on board your suggestions and agree that the papers you refer to are becoming a benchmark for analysing energy expenditure datasets, however of course there still are many papers in *NPG* journals, and others, that do not publish EE data in this format. Nevertheless, we wish to demonstrate the robustness of our data in the best way possible, therefore have reanalysed the EE data according to PMID: 22205519 and present the findings in Figure 4 and Supplemental Figure S4. These data clearly demonstrate using ANCOVA that there is a genotype effect in female mice that is independent of the changes in fat mass and lean mass. This is clear in both the NC and HFD fed mice and supports our other datasets that the female mice demonstrate a more striking phenotype in response to adipose specific deletion of Trim28.

With regards to food intake, unfortunately it was difficult to get accurate or meaningful measurements of food intake as the mice were housed as littermates, thus each cage had a mix of WT and KO together. In addition, during the CLAMS studies both the WT & KO mice crumbled their food, making it difficult to get reliable assessments of food intake.

7. The RER data is not interpreted. First RER values variate between 0.7 (pure fatty acid oxidation) and 1 (pure glucose oxidation), the graphs of RER should be plotted from a higher value than 0 in order to appreciate better the effect size.

It is worth mentioning that this mouse model may have a metabolic inflexibility phenotype, potentially as a result of an impaired lipolysis (as suggested by the p-HSL blot in Fig. 6 or the lipidomic data). However, the reduced RER may suggest the opposite.

We now present RER data for NC and HFD fed mice in Figure 4A-D and 4I-L. Of course, the calculation of EE also relies heavily on the RER calculations, and thus these data are just a different way of interpreting the RER output. We also agree that it may be difficult to delineate differences with the y-axis starting at 0, however it was our belief that it is good practice to show the full axis, as quite often data is presented where the y-axis does not start at 0 in attempts to accentuate minor differences. Nevertheless, out of respect for the reviewer's request, we have adjusted the scale on the y-axis where appropriate in the main figures, to better demonstrate where differences exist. We have also included altered scales for Figures 4B, D, J and L below, so the reviewer can see the data in its more accentuated format.

Can the authors provide fed and fasted FFA and TG levels to further explore this aspect?

This is something we would have liked to have done, and indeed had considered to be of potential interest. However, in light of our many new data sets, we believe these data would provide only an incremental increase in evidence to support our current claims, and thus it is our opinion that these studies are outweighed by the commitment of resources and time that would be required to complete them. However, as described below relating to our lipolysis data, we have performed a number of new in vitro and in vivo based experiments to demonstrate more robust and consistent findings relating to alterations in the lipolysis pathways in the setting of Trim28 depletion. See Figures 6 and 7.

Additionally, markers of lipid uptake by adipose tissue should be also presented. Recent publications have shown the importance of adipose tissue in the regulation of metabolic flexibility (PMID: 29789270 and PMID: 30134163).

Using RNA-sequencing data we now have the capacity to investigate all differentially expressed genes in both male and female adi-KO mice compared to control mice. This includes genes involved in FAO, lipid uptake and synthesis. Indeed, Figure 7 demonstrates that adi-KO mice have global alterations in pathways associated with fatty acid metabolism, adipogenesis and differentiation – which are suggestive of some of these genes being altered. Moreover, when investigating specific genes of interest, we demonstrate differential expression of LPL, Pck1 and several Elovl genes, however we do not see a major alteration in genes that would be supportive of a metabolic inflexibility phenotype. We do however observe substantial alterations to the gene Olr1, which has been demonstrated to be strongly associated with lipid binding and cholesterol levels in adipocytes.

8. The authors claim that Trim28 deletion increases BW without affecting glucose metabolism. The GTT provided in Fig. 2K-L however suggests the opposite. If the authors want to make such strong

statement, they should assess glucose homeostasis and insulin sensitivity using state of the art techniques such as hyperinsulinemic-euglycemic clamp.

We thank the reviewer for this observation, and they are indeed correct that from this dataset, glucose tolerance *appears* to be mildly impaired. However, as now mentioned in detail the text (pp 10, lines 206-221), these GTTs were dosed according to total body mass. There are three ways that mice can be dosed for these studies; 1. According to total body weight, 2. According to lean mass and 3. A standard dose across the board (similar to what is seen in the clinic). There are pro's and con's for each approach, but the biggest consideration should be that lean mass is the primary source for glucose uptake and thus the weight of lean mass will influence the clearance of glucose the most. Our Trim28 adi-KO mice have greater body weights because of increased fat mass, but lean mass is equivalent between genotypes. Therefore, when we dose these mice according to body weight, the KO mice receive more glucose, meaning they have a greater load to clear. Because their lean mass is similar, it takes these mice longer to clear the glucose and thus this presents as "glucose intolerance" in the data. This is a common problem in GTT experiments that is often mis-interpreted in many published studies. Thus, when we re-performed the GTT according to lean mass, meaning now all mice received a more equivalent dose of glucose, the apparent impairment disappeared (see Supplementary Figures S3A & S3B).

Furthermore, whilst it is true that the euglycemic hyperinsulinemic clamp is the gold standard for measuring insulin resistance, we in fact see very little evidence of major differences in insulin resistance in this model, thus it would be unethical for us to perform such burdensome experiments on so many mice, to demonstrate what is highly likely to result in an inconclusive outcome.

9. Why the adipose tissue may be different is unclear at this stage. Depots mass, histology of the tissues, notably assessing adipocyte size, may help as well as RNAseq to define the pathways altered in Trim28 KO may guide the authors toward a potential mechanism that could explain their phenotype.

We agree with the reviewer that our mechanistic understanding for what is driving this phenotype was limited in our initial submission. As such, we have now performed an in-depth and robust (n=6/group) RNA-sequencing experiment from white adipose tissue of both males and females WT and KO mice.

As detailed in Figure 7, these data detail striking effects of Trim28 depletion of lipid metabolism pathways, confirming many of our previous observations using qPCR, but also identifying a number of highly interesting targets that help explain this phenotype, including the sexual dimorphic response. Most notably, were the prominent changes in expression of the transcriptional regulator, Klf14. Several robust and highly reputable groups have demonstrated that Klf14 is a bona fide regulator of adipose tissue development and a genetic regulator of adiposity, that acts in a sexual dimorphic manner favouring effects in females. However, previous studies have not detailed what regulates the expression of Klf14. Thus, our data is the first to report a transcriptional regulator of Klf14 expression, both implicating Trim28 as an important component of this network and providing a highly plausible explanation for the phenotype in our Trim28 adi-KO mouse models.

10. The lipidomic data show that Trim28-KO mice have alteration in chain length and desaturation of their TG that are somehow mirrored in the serum. Can the authors provide gene expression data

regarding more elongases than Elovl3 (which is a browning marker) and of the fatty acid desaturases (SCDs, FADS).

RNA-seq analysis now demonstrates that several Elovl family members are regulated in adi-KO WAT, as shown in Figure 7K. These data provide further evidence that Trim28 regulates pathways that lead specific alterations in the adipocyte TG pool, consistent with our lipidomics analysis.

11. It is quite surprising to observe such an increase in Elovl3 in ScWAT, this data can support a browning phenotype of the WAT and a potential increased SNS tone. Given that adiponectin is also expressed in BAT and therefore Trim28 must be deleted in BAT adipocytes, it may have a functional impact. Can the authors provide gene expression analyses of the BAT, ScWAT and gWAT of the thermogenic markers (e.g. UCP1, Dio2, Elovl3, etc.)?

We thank the reviewer for bringing this to our attention, and we are indeed aware that these findings are consistent with a browning phenotype. We demonstrate that Elovl3 is markedly increased in gonadal WAT, sub cutaneous WAT and BAT (Supp Fig S8B). However, we do not believe this is a result of browning, because other bona fide markers of browning are not significantly altered as demonstrated by qPCR and RNA-seq analysis. See Figures 7E & 7F.

The impaired lipolysis is so far the clearest result of the study and should be supported by a proper lipolysis experiment in AT from female KO mice. Are the mice resistant to adrenergic agonist? Is this phenotype also found in KO male? As for comment 7, fed and fasted FFA and TG levels will help to confirm the potential lipolytic phenotype.

Thank you for these suggestions, and these were experiments we had considered also. However, because we see only mild alterations in plasma lipids in these models, we believe that specific adrenergic activation experiments in mice are likely to result in moderate or inconclusive outcomes. Therefore, to eliminate this variability and the complexities of the in vivo milieu, we have since performed lipolysis studies in trim28-depleted 3T3-L1 adipocytes, using shRNAs. In these studies, we demonstrate that glycerol secretion is reduced in Trim28 knock down cells both basally and in the presence of isoproterenol (adrenergic agent), and that equivalent alterations in pHS1 signalling were also observed in these cells, as was observed in vivo. Thus these data validate that deletion of Trim28 leads to robust alterations in adipocyte lipolysis. See Figures 6G & 6H.

12. The authors claimed the obese KO mice to be healthier. Are the mice protected from liver steatosis ?

Our group has significant experience in using lipidomics data to evaluate liver lipid abundance and physiology associated with hepatosteatosis in mice (see Parker et al., Nature, 2019). The lipidomics data performed on the liver in this study suggest that the abundance of many TG species, particularly the shorter chain species, which are a cardinal feature of steatosis, were reduced compared to WT mice. This is also consistent with the fact that we observe improved fasting blood glucose levels, which is primarily regulated by the liver via hepatic glucose production (HGP). This is also consistent with the alterations we observed in Akt phosphorylation in Supp Fig s1J.

Minor comments:

1. Acaca is not a "gene involved in adipose tissue lipolysis", it is the first step of de novo lipogenesis

and ACC is actually a target of AMPK. AMPK phosphorylates ACC in condition of energy deprivation to reduce DNL and increase FAO. Additionally, there is not a reduction in ACC phosphorylation but a general reduction in ACC protein. The authors should amend their manuscript to avoid any confusion regarding this pathway.

We apologise for this oversight and have now corrected this in the manuscript. See lines 453-455.

2. The authors should present GTT results from male and female in two different graph to facilitate the interpretation.

We now present the data separated by sex as suggested by the reviewer. See Figures 1 & 2.

3. The n number is wrong in the legend of figure 6. The n numbers are actually different from one panel to another.

N numbers have now been corrected.

Reviewers' comments:

Reviewer #1 (Remarks to the Author):

The authors addressed all my concerns

Reviewer #2 (Remarks to the Author):

We appreciate the effort of the authors substantially revising this manuscript. While the authors have improved their manuscript, we believe the work is still premature in its current form and some issues remain to be addressed.

The RNAseq analysis of the WAT of Trim28-KO highlighted Or1 and Klf14 as potential candidates by which Trim28 regulates adiposity. While these results are of potential interest they have not been satisfactorily validated and elucidation of their role in triggering Trim28 KO phenotype would strongly reinforce the conclusions. Also we feel the shape of the volcano plot presented in Fig. 7G is very unusual, and for this reason the results of the entire dataset of the RNAseq analysis would be appreciated.

Moreover, major shortcomings are remaining and in our opinion it would be important to be satisfactorily addressed. The authors provided incomplete data to answer some of our comments.

Comment 1: We appreciate the authors answer but we feel the question regarding other cells expressing Trim28 has not been satisfactorily addressed. In fact Trim28 is notably expressed in macrophages and its inhibition enhanced the inflammatory response to LPS (PMID22995936). The relative expression of Trim28 within the different cell types of the WAT is important and we would encourage the authors to look into using publically available datasets if it were impossible to produce this information.

Comment 2: The authors addressed this point satisfactorily.

Comment 3: The authors showed that the insulin levels were not different among the genotypes but claimed that Trim28-KO mice might be more insulin sensitive. Increased insulin signalling in AT could partly explain that Trim28-KO mice show an increased adiposity and inhibition of lipolysis. While the authors presented AKT phosphorylation in the liver, insulin signalling in the adipose tissue was not been investigated. We insist measuring FFA levels, which can be used as a proxy for adipose tissue insulin sensitivity, given that insulin is a well describe brake for adipose tissue lipolysis. Also we wil encourage the quantification of the liver pAKT blot presented in Fig. S1J to support the authors conclusion.

Comment 4: This point was addressed satisfactorily

Comment 5: Statistical approach was satisfactorily clarified

Comment 6: The data showed in figure 1F does not show in our opinion a “robust alteration in energy expenditure” in Trim28KO as claimed by the authors. It is unclear how this ANCOVA had been performed. It seems that the authors exclusively tested the significant differences between the slope rather than comparing of the Y intercept. Hence asking the question: are my mice having an increase EE independently of changes in BW?

It is a pity that the cohort used to perform the new GTT was not used to determine the food intake. Without this data the energy balance phenotyping is incomplete and inconclusive.

Comment 7: We appreciate the authors followed the advice of changing their scale of the RER. While we agree that full scales are usually of good practice in science, when a variable is not expected to go below a certain point, hereby 0.7 for the RER - which correspond to a state of pure fatty acid oxidation -, it makes no sense to plot the data from 0. This comment was surprising since it was not an issue in the first version in their manuscript where Y axis not starting from 0 were presented in Fig. 1A, S1C and D. The same apply in this version of the manuscript.

We are not sure we understand what is the problem of measuring fed and fasted FFA and TG in their model. Given the importance of impaired lipolysis of the AT for the phenotype, we felt this was an easy relatively cheap analysis that could strength this claim. This also relates to comment 11, indicating that there are many ways to induce lipolysis in vivo, one is using a beta-3 adrenergic agonist (such as CL-316,243), but fasting or cold exposure would also trigger a robust lipolytic response. We feel these measurements are important and within the capabilities of the authors.

Comment 8: The new GTT and ITT reinforce the conclusions of this manuscript. Conversely the ITT in Fig. S3C is unusual as the blood glucose of their animals go down, then up, then down, which is difficult to interpret.

Comment 9: The transcriptomic analysis of the WAT in a real add-value to the manuscript by identifying potential targets that obviously require further validation.

Comment 10: The authors satisfactorily addressed this comment.

Comment 11: The study of FFA and TG in different status of adrenergic stimuli (i.e. fasting, feeding, cold exposure, b3-agonism, etc.) is of great importance to support the main message of the manuscript.

Comment 12 and minor comments: The authors satisfactorily addressed these comments.

Please find our second response to the reviewers' comments below.

Reviewer #1 (Remarks to the Author):

The authors addressed all my concerns

Reviewer #2 (Remarks to the Author):

We appreciate the effort of the authors substantially revising this manuscript. While the authors have improved their manuscript, we believe the work is still premature in its current form and some issues remain to be addressed.

The RNAseq analysis of the WAT of Trim28-KO highlighted *Orl1* and *Klf14* as potential candidates by which Trim28 regulates adiposity. While these results are of potential interest they have not been satisfactorily validated and elucidation of their role in triggering Trim28 KO phenotype would strongly reinforce the conclusions. Also we feel the shape of the volcano plot presented in Fig. 7G is very unusual, and for this reason the results of the entire dataset of the RNAseq analysis would be appreciated.

We appreciate that the reviewer finds these results of interest, and as indicated in comment 9 they recognize these additions as a “real add-value” to the manuscript. It is our opinion that these data sufficiently answer the original query, which was that our initial submission “lacked mechanistic insight to explain the observed phenotype”. Our new RNA-seq data provides substantial mechanistic insight that provides a very plausible explanation for the explained phenotype.

Nevertheless, in the interests of continuing to strengthen the findings in our study, we now provide additional evidence to support these claims. We present new data from *in silico* modelling analysis of deposited protein-protein interaction (PPI) datasets, which suggests that Trim28 and KLF14 are likely to be co-members of a shared transcriptional repressor complex, facilitated by an interaction with Sin3a (Supplementary Figure S10). Both KLF14 and Trim28 directly interact with Sin3a – a critical component of HDAC1 containing repressor complexes. Thus, these data provide evidence that loss of Trim28 in adipose tissue may alter activity of repressor complexes harbouring KLF14/Sin3a, which in turn would impact HDAC1 activity.

Regarding the plot in 7G, we agree that the shape of the volcano plot is unusual with the skew of data points demonstrating a bulk increase in expression of many genes at or just above the threshold of significance. We have performed the analysis in several different ways to demonstrate that this is not due to incorrect analysis from gene counts or FDR correction (see other plots below). Indeed, if we plot p-value or q-value we observe the same pattern. Instead, we believe this is due to a bulk upregulation of several co-regulated pathways, all of which having varying degrees of differential expression but which display natural variation in the replicates so as to remain on the border of FDR significance. A Spearman's correlation analysis has confirmed a high degree of co-regulation amongst networks (data not shown).

We have included the list of differentially expressed genes (DEGs) in Supp Table S2, and will deposit the full datasets on FigShare.

Volcano plot of RNA-seq data analysed for differential expression using a different software package to that shown in figure 7G. Note the same “skewed” data points are observed to the right of the plot.

Moreover, major shortcomings are remaining and in our opinion it would be important to be satisfactorily addressed. The authors provided incomplete data to answer some of our comments.

Comment 1: We appreciate the authors answer but we feel the question regarding other cells expressing Trim28 has not been satisfactorily addressed. In fact Trim28 is notably expressed in macrophages and its inhibition enhanced the inflammatory response to LPS (PMID22995936). The relative expression of Trim28 within the different cell types of the WAT is important and we would encourage the authors to look into using publically available datasets if it were impossible to produce this information.

We remain confused as to what the reviewer is requesting in this section. We are aware that Trim28 is expressed in many cells types, several of which will be present in the adipose depots. Indeed, the residual Trim28 expression in KO WAT is likely to originate from these non-adipocyte, Trim28 replete cell types. However, we are generating an adipocyte specific KO of Trim28, so we are uncertain as to how the expression of Trim28 in other non-adipocyte cell types would be of relevance to our phenotype. Perhaps the reviewer is suggesting that our model also results in Trim28 deletion in macrophages? This is unlikely, because if this were the case then we might have expected to observe changes in inflammatory markers or macrophage gene signatures in our RNA-seq - however we did not.

We believe that an important finding from our study is that deletion of Trim28 in committed adipocytes promotes an increased adiposity and alterations in adipocyte lipid metabolism. Our data supports this conclusion, which is in contrast to previous studies that suggested committed cell types were unlikely to drive the phenotype.

Comment 2: The authors addressed this point satisfactorily.

Comment 3: The authors showed that the insulin levels were not different among the genotypes but claimed that Trim28-KO mice might be more insulin sensitive. Increased insulin signalling in AT could partly explain that Trim28-KO mice show an increased adiposity and inhibition of lipolysis. While the authors presented AKT phosphorylation in the liver, insulin signalling in the adipose tissue was not been investigated. We insist measuring FFA levels, which can be used as a proxy for adipose tissue insulin sensitivity, given that insulin is a well describe brake for adipose tissue lipolysis. Also we wil encourage the quantification of the liver pAKT blot presented in Fig. S1J to support the authors conclusion.

We find it unusual for a reviewer to *insist* on measuring an output that is, at their own admission, a “proxy” for the pathway of interest. Indeed, fatty acid abundance in the plasma can be regulated under many conditions and by several tissues, only one of which is insulin action in the adipose tissue. We already provide a direct measure of lipolysis activation in adipose tissue - in vivo phosphorylation of HSL (which was also confirmed in an in vitro model). Nevertheless, we have now included measurements of total NEFA and specific low abundance species of FFA in plasma of adi-WT and adi-KO mice (Supplementary Figures 8C&D). These data demonstrate a trend for reduced plasma FFAs in Trim28-KO mice, but as expected these effects are mild as a result of the many tissues that contribute to plasma FFA abundance. Nevertheless, these data are consistent with our overall hypothesis that Trim28-KO leads to an inhibition of lipolysis signalling in WAT.

Quantification for the blots in S1J is now shown in Figure S1K.

Comment 4: This point was addressed satisfactorily

Comment 5: Statistical approach was satisfactorily clarified

Comment 6: The data showed in figure 1F does not show in our opinion a “robust alteration in energy expenditure” in Trim28KO as claimed by the authors. It is unclear how this ANCOVA had been performed. It seems that the authors exclusively tested the significant differences between the slope rather than comparing of the Y intercept. Hence asking the question: are my mice having an increase EE independently of changes in BW?

Our data demonstrates that energy expenditure (EE) is altered in Trim28 adi-KO female mice independently of BW and adiposity, indicating that the adiposity phenotype itself is not responsible for these differences. This altered EE is likely to explain in part, the observed phenotype - which was the main point of these analyses and one of the original concerns of the reviewer. We have now changed our wording in the manuscript to reflect this interpretation.

It is a pity that the cohort used to perform the new GTT was not used to determine

the food intake. Without this data the energy balance phenotyping is incomplete and inconclusive.

We are sorry but there may be some confusion. We did not run another cohort for the revision to perform a new GTT. The GTT the reviewer is referring to here has always been part of the original and resubmitted manuscripts, just that we discussed the results of these data in greater detail in the response to reviewers.

Comment 7: We appreciate the authors followed the advice of changing their scale of the RER. While we agree that full scales are usually of good practice in science, when a variable is not expected to go below a certain point, hereby 0.7 for the RER - which correspond to a state of pure fatty acid oxidation -, it makes no sense to plot the data from 0. This comment was surprising since it was not an issue in the first version in their manuscript where Y axis not starting from 0 were presented in Fig. 1A, S1C and D. The same apply in this version of the manuscript.

We are not sure we understand what is the problem of measuring fed and fasted FFA and TG in their model. Given the importance of impaired lipolysis of the AT for the phenotype, we felt this was an easy relatively cheap analysis that could strength this claim. This also relates to comment 11, indicating that there are many ways to induce lipolysis in vivo, one is using a beta-3 adrenergic agonist (such as CL-316,243), but fasting or cold exposure would also trigger a robust lipolytic response. We feel these measurements are important and within the capabilities of the authors.

We now provide measures of fasting plasma FFA levels in Supplementary Figures 8A and 8B. As discussed above, these data suggest a trend for reduced FFA levels in plasma, but given the several tissues that contribute to plasma FA abundance, this does not reach significance.

Comment 8: The new GTT and ITT reinforce the conclusions of this manuscript. Conversely the ITT in Fig. S3C is unusual as the blood glucose of their animals go down, then up, then down, which is difficult to interpret.

It should be noted that most groups only present ITT plasma glucose data up to 60 minutes post injection, whilst our data is presented out to 120 minutes – consistent with GTT data. This is often justified by others because the circulating half-life of insulin is less than 1 hour. So, yes whilst our ITT results may look “unusual”, others have likely also thought their data looked unusual and perhaps cut their graph short, hence we don’t often see the full 120 minutes.

Nevertheless, there is a physiological explanation. During an ITT the exogenous delivery of insulin rapidly and expectedly reduces blood glucose (the first drop in glucose levels). This in turn can result in compensatory increases in hepatic glucose production in an effort to protect the animal from hypoglycemia (the “rebound” in glucose). This rebound evokes a secondary rise in insulin, this time as a result of secretion from the pancreas, which again reduces glucose (the second drop in glucose).

Comment 9: The transcriptomic analysis of the WAT in a real add-value to the manuscript by identifying potential targets that obviously require further validation.

Comment 10: The authors satisfactorily addressed this comment.

Comment 11: The study of FFA and TG in different status of adrenergic stimuli (i.e. fasting, feeding, cold exposure, β 3-agonism, etc.) is of great importance to support the main message of the manuscript.

We have now provided data to demonstrate that fasting FFA levels are reduced in Trim28 adi-KO mice compared to WT mice (particularly in female mice). This provides a further read out of impaired lipolysis in these mice, supporting the main conclusions of the manuscript.

Comment 12 and minor comments: The authors satisfactorily addressed these comments.

REVIEWER COMMENTS

Reviewer #2 (Remarks to the Author):

We thank the authors for their efforts in improving their manuscript. While some concerns have been satisfactorily addressed, the authors still provided incomplete data to answer some of our comments. The impact of Trim28 deletion on AT physiology remains to be elucidated.

1. One significant comment is related to adipose tissue insulin signaling. The authors have measured in this revised version the FFA levels, which partly support the results showing decreased lipolysis (the authors should better define a “trend”, what is the p-value? The fold-change?). However, the authors did not provide any data regarding insulin signaling in the adipose tissue of Trim28-KO mice as it was indicated in comment 3. Increased insulin signalling could partly explain that Trim28-KO mice show increased adiposity and inhibition of lipolysis. It is not clear how Trim28 regulates adiposity and metabolic functions, and investigating insulin signalling would provide some insights regarding the mechanisms underlying the phenotype of the KO mice.

2. The analysis of the energy expenditure remains obscure. The presented plots clearly show that EE is not different between the WT and the KO, and the term “altered” used by the authors is not conclusive. Furthermore, the authors are still claiming that the EE is increased in female KO mice (in the title of the paragraph page 11, line 234), which does not seem real.

3. The absence of food intake analysis remains an important caveat of this manuscript, and this limitation should be acknowledged in the discussion of the manuscript.

4. The FFA level data is not providing evidence regarding the fact that Trim28-KO mice have impaired lipolysis, at least in response to fasting. First, as mentioned above, the authors mention the decrease as a trend without details. Also, in chow-fed animals, the RER of the KO mice is lower during the light phase, when AT lipolysis is the more active, suggesting that Trim28-KO mice are using more FA than glucose as substrate for oxidation, therefore going against an impaired lipolysis phenotype, at least in the fasted state. RER is also lower during the dark phase, thus suggesting that Trim28-KO mice could have: 1) an impaired inhibition of lipolysis during the fed state (as a result of AT insulin resistance, as aforementioned) and/or 2) an impaired adipose tissue lipid uptake and/or 3) a reduced food intake. Overall, we encourage the authors to revise the analysis of their data. In our opinion, Trim28-KO mice have a dysfunctional AT that is incapable of doing its primary functions of storing energy in fed states and releasing it during fast. We believe the manuscript will gain clarity regarding the role of Trim28 in the regulation of AT physiology if the author could discuss all these aspects.

5. The explanation of the ITT data by the authors (comment 8) is surprising. It is not easy to justify an unusual result stating that others may have hidden it. The second decrease is still very surprising since it occurs while the blood glucose did not even reach the basal levels and should, therefore, not trigger a secondary insulin response. Given the shape of the curve of this ITT, the AUC analysis is unlikely to reflect the insulin sensitivity of these animals.

Minor comment: It is challenging to navigate through the manuscript without the figure number written in the figures.

Reviewer 2:

We thank the authors for their efforts in improving their manuscript. While some concerns have been satisfactorily addressed, the authors still provided incomplete data to answer some of our comments. The impact of Trim28 deletion on AT physiology remains to be elucidated.

We thank reviewer 2 for their experimental recommendations. We have performed additional experiments in regards to AT physiology, and made changes in the text where appropriate in response to reviewer 2's critique.

1. One significant comment is related to adipose tissue insulin signaling. The authors have measured in this revised version the FFA levels, which partly support the results showing decreased lipolysis (the authors should better define a "trend", what is the p-value? The fold-change?).

We have now included the p-values and percent decrease for NEFA/FFA into the text (line 357) and figures (Figure S8) for plasma FFA measurements.

However, the authors did not provide any data regarding insulin signaling in the adipose tissue of Trim28-KO mice as it was indicated in comment 3. Increased insulin signalling could partly explain that Trim28-KO mice show increased adiposity and inhibition of lipolysis. It is not clear how Trim28 regulates adiposity and metabolic functions, and investigating insulin signalling would provide some insights regarding the mechanisms underlying the phenotype of the KO mice.

We have now conducted experiments in Figure 6 to analyze critical components of the insulin signaling pathway in adipose tissue of WT and Trim28 adi-KO mice. These experiments indicate that the alterations to lipolysis in Trim28 adi-KO mice are independent of insulin signaling in adipose tissue. This was demonstrated using Western blotting in WAT lysates (Figure 6D-6F), which shows no significant difference in the level of Akt or GSK3beta phosphorylation between WT and KO animals. This suggests that adipose specific deletion of Trim28 regulates lipolysis through mechanisms that are not directly related to canonical insulin signaling, and is thus unlikely to be related to insulin resistance. This is consistent with our cell culture data in 3T3-L1 adipocytes, which demonstrates a similar alteration in phosphorylation of HSL and reduced glycerol release in Trim28 knock-down cells, in the absence of any exogenous insulin effect.

2. The analysis of the energy expenditure remains obscure. The presented plots clearly show that EE is not different between the WT and the KO, and the term "altered" used by the authors is not conclusive. Furthermore, the authors are still claiming that the EE is increased in female KO mice (in the title of the paragraph page 11, line 234), which does not seem real.

In our original submission we demonstrated that adi-KO mice had a higher EE compared to WT mice after adjustment for body weight and/or lean mass (most notably in females). We appreciate that this analysis can be confounded by a number of factors, thus we subsequently perform an analysis of covariance (ANCOVA) at the request of the reviewer and as described in the paper by Tschop et al (Nat Methods, 2013). ANCOVA determines if the dependent variable (EE) is influenced by an independent variable (genotype) after controlling for covariables (body weight/lean mass).

The ANCOVA results in our analysis indicate that there is a significant genotype effect on EE that is independent of covariables including body mass and lean mass. This supports our conclusion that adi-KO mice have an altered EE compared to WT mice, and we propose that this finding - in combination with our original analysis of the CLAMS data – supports our hypothesis that female adi-KO mice have increased EE. This is readily observable in Figure 4N, where mice that have an equivalent lean mass at the lower end of the spectrum (~16g of lean mass), demonstrate two distinct groupings between WT and adi-KO mice (red vs black dots). The linearity of these data as a function of lean mass are also different, providing further evidence of an effect of the adi-KO genotype on peripheral EE - that is not dependent on lean mass. We hypothesize that these data

indicate that as KO animals get larger, lean mass is becoming unable to compensate appropriately to increase EE, which may lead to an eventual negative metabolic state.

We now incorporate aspects of these data/discussions in the manuscript (page 11&12).

3. The absence of food intake analysis remains an important caveat of this manuscript, and this limitation should be acknowledged in the discussion of the manuscript.

We agree with reviewer 2 that the measurement of food intake would have been an additional piece of evidence to support our findings. However, given that previous studies investigating the metabolic role of Trim28, including whole body haploinsufficient mice in the study by Dalgaard et al., showed no change in food intake we didn't anticipate that this would be a contributing factor. We now discuss the lack of an effect on food intake from these other studies in the manuscript (line 451), and acknowledge that our findings should be interpreted with this caveat in mind.

4. The FFA level data is not providing evidence regarding the fact that Trim28-KO mice have impaired lipolysis, at least in response to fasting. First, as mentioned above, the authors mention the decrease as a trend without details. Also, in chow-fed animals, the RER of the KO mice is lower during the light phase, when AT lipolysis is the more active, suggesting that Trim28-KO mice are using more FA than glucose as substrate for oxidation, therefore going against an impaired lipolysis phenotype, at least in the fasted state. RER is also lower during the dark phase, thus suggesting that Trim28-KO mice could have: 1) an impaired inhibition of lipolysis during the fed state (as a result of AT insulin resistance, as aforementioned) and/or 2) an impaired adipose tissue lipid uptake and/or 3) a reduced food intake. Overall, we encourage the authors to revise the analysis of their data. In our opinion, Trim28-KO mice have a dysfunctional AT that is incapable of doing its primary functions of storing energy in fed states and releasing it during fast. We believe the manuscript will gain clarity regarding the role of Trim28 in the regulation of AT physiology if the author could discuss all these aspects.

We thank reviewer 2 for their efforts in trying to help us understand the complex phenotype of this model – however in light of the new data provided in this revision, we don't believe the above interpretations are necessarily accurate.

1. Data from adipose tissue lysates shows substantial and robust alterations in HSL phosphorylation, yet there is little evidence to suggest a reduced activation of the insulin signaling pathway.
2. Previous findings from whole body, brain specific and liver specific Trim28 KO mice do not provide any evidence for an alteration in food intake
3. Our existing and extensive lipidomics, transcriptomics and animal phenotyping data provides no evidence for a reduced lipid uptake in adipose tissue. Most notably, the animals are fatter – which is in contrast to what would be expected if lipid uptake was impaired.

However, we do agree that there may be some aspect of adipocyte metabolic inflexibility in our model, which we already allude to in the manuscript text. However, the precise mechanisms behind this and the impact it has on whole body metabolism (e.g. as reflected in RER etc) are not completely understood. However, these likely relate to the vast transcriptional networks that are altered in WAT in the absence of Trim28, including those regulated by KLF14.

5. The explanation of the ITT data by the authors (comment 8) is surprising. It is not easy to justify an unusual result stating that others may have hidden it. The second decrease is still very surprising since it occurs while the blood glucose did not even reach the basal levels and should, therefore, not trigger a secondary insulin response. Given the shape of the curve of this ITT, the AUC analysis is unlikely to reflect the insulin sensitivity of these animals.

The half life of insulin in mouse circulation is less than 60 minutes, and thus any effect of injected insulin have subsided by that time point. When we analyze the change in FBG before or at 60mins following a standardized exogenous insulin bolus there is no difference between WT and KO animals.

The aim of performing the ITT is to provide additional information in relation to the animals' metabolic status, which are interpreted in conjunction with other readouts such as GTT. When we consider this lack of effect in the ITT, together with GTT data and new adipose tissue insulin signaling data, we interpret this to indicate that Trim28 adi-KO mice do not have a significant impairment in whole body insulin sensitivity.

Minor comment: It is challenging to navigate through the manuscript without the figure number written in the figures.

We apologize for this oversight, and have now amended the manuscript and figures to align with the appropriate references to figures.

REVIEWER COMMENTS

Reviewer #2 (Remarks to the Author):

Comments: Deletion of Trim28 in Committed Adipocytes Promotes Obesity but Preserves Metabolic Health

The authors improved the manuscript and addressed some of our comments. However, the interpretation of some elements of the study remain unclear and need to be readdressed. It remains unclear which aspect of lipolysis is impaired. Looking at the data it seems that this model relates more to impaired metabolic flexibility arising from the adipose tissue rather than a specific alteration in lipolysis. For instance the reduction of HSL phosphorylation is associated with a reduction in ACC. This association coupling lipolysis and lipogenesis, namely the balance between storage and release of energy, could be the root of the phenotype. The data are still complex to interpret due to the discrepancy of the results:

1. The authors showed in Figure 1 that Trim28 expression is decreased during differentiation suggesting a role of Trim28 regulation in the last stage of adipogenesis, enabling lipid storage. This claim is supported by the literature but not by their results, presented in supplementary figure 8, showing no effect of trim 28 deletion in adipocyte lipid storage. Moreover, no gene expression is provided to inform on the differentiation capacity of the trim28 depleted cells.
2. The increase of EE remains unconvincing, and it is not supported given the absence of modification of the thermogenic markers. This is more supportive of dysfunctional AT. The decrease in RER is in favour of increased fatty acid oxidation which is in agreement with increased lipolysis. However, the authors described impaired lipolysis in adipocytes depleted for trim28. RER or EE modification in trim28 depleted mice is not discussed in the manuscript.

We think that the data presented in the manuscript support that Trim28 sets adipose tissue metabolic flexibility and function. As such, it is difficult to state in the title that these animals present a "preserved metabolic health" since AT function (storage and release of E) seems compromised. Overall, in our opinion, the manuscript would benefit from a reshaping in the interpretation to fit better a phenotype of metabolic inflexibility rather than focusing exclusively on a mechanistically unclear alteration in lipolysis and unexplained obesity phenotype.

Reviewer #2 (Remarks to the Author):

Comments: Deletion of Trim28 in Committed Adipocytes Promotes Obesity but Preserves Metabolic Health

The authors improved the manuscript and addressed some of our comments. However, the interpretation of some elements of the study remain unclear and need to be readdressed. It remains unclear which aspect of lipolysis is impaired. Looking at the data it seems that this model relates more to impaired metabolic flexibility arising from the adipose tissue rather than a specific alteration in lipolysis. For instance the reduction of HSL phosphorylation is associated with a reduction in ACC. This association coupling lipolysis and lipogenesis, namely the balance between storage and release of energy, could be the root of the phenotype. The data are still complex to interpret due to the discrepancy of the results:

1. The authors showed in Figure 1 that Trim28 expression is decreased during differentiation suggesting a role of Trim28 regulation in the last stage of adipogenesis, enabling lipid storage. This claim is supported by the literature but not by their results, presented in supplementary figure 8, showing no effect of trim 28 deletion in adipocyte lipid storage. Moreover, no gene expression is provided to inform on the differentiation capacity of the trim28 depleted cells.

Indeed, we do show reductions in Trim28 protein abundance during differentiation of 3T3-L1 adipocytes, which supports our hypothesis that Trim28 is associated with lipid storage programs in adipocytes. However, we are unsure as to the data the reviewer is referring to in Figure S8. We presume Figure S8E (brightfield images). These images demonstrate that the level of differentiation between Trim28 depleted and replete cells, is not vastly different on a morphological level. This of course is not a quantitative measure of lipid burden, but certainly demonstrates that there is no evidence for reduced lipid loading (which was our primary concern when showing robust differences in HSL phosphorylation). However, to appease the reviewers concerns we have now provided further western blotting data (instead of gene expression) which demonstrates that there is no impairment in the expression of pro-adipogenic proteins PPARgamma and C/EBPalpha in Trim28 depleted 3T3-L1 cells (see updated Supplementary Figure 8D). Below, we also show a time course brightfield of these cells (day 3 and day 6 – where day 10 is shown in Supp Fig S8E), to further demonstrate the consistency in differentiation between control and Trim28 depleted cells. Whilst this data might appear unexpected given our findings on lipolysis both in mice and in these cells, it is important to consider cultured cells do not get the same exposures to nutrient and diurnal flux, which likely minimises the impact of any cellular alterations.

2. The increase of EE remains unconvincing, and it is not supported given the absence of modification of the thermogenic markers. This is more supportive of dysfunctional AT. The decrease in RER is in favour of increased fatty acid oxidation which is in agreement with increased lipolysis. However, the authors described impaired lipolysis in adipocytes depleted for trim28.

We appreciate the reviewer's thoughts in regards to interpreting the EE and RER data. This is a complex phenotype and therefore we welcome insights from other experts in the field. We are a little surprised that the reviewer is of the opinion that the EE and RER are unconvincing, particularly when they are statistically significant ($p < 0.018$ and $p < 0.04$ respectively) according to the ANCOVA protocol. As such, we respectively disagree that they are unconvincing from this perspective. However, we do appreciate that the normal function of white adipose tissue may be altered in this model, perhaps relating to an inflexibility phenotype as suggested. **We have now included some discussion in the revised manuscript to acknowledge this potential mechanism.**

The alterations in RER between adi-KO and WT mice are small, and suggestive that a mixed substrate is being utilized for energy production (RER = ~ 0.85) in KO mice, where the WT mice are preferentially utilising glucose (RER = ~ 0.95). Thus, the inhibition in adipocyte lipolysis could easily be compensated for by other tissues. This likely involves the liver, which we showed had a reduced TG content in KOs, supportive of the notion that the liver may buffer FA levels in the blood of KO mice. This would also provide an explanation for the lack of insulin resistance in the liver in the heavier adi-KO mice, as FAs and TGs were fluxing more efficiently through the liver.

RER or EE modification in trim28 depleted mice is not discussed in the manuscript. We discuss these parameters on page 11 and 12.

We think that the data presented in the manuscript support that Trim28 sets adipose tissue metabolic flexibility and function. As such, it is, difficult to state in the title that these animals present a "preserved metabolic health" since AT function (storage and release of E) seems compromised. Overall, in our opinion, the manuscript would benefit from a reshaping in the interpretation to fit better a phenotype of metabolic inflexibility rather than focusing exclusively on a mechanistically unclear alteration in lipolysis and unexplained obesity phenotype.

We take on board the comments from the reviewer suggesting a refocus of our main points, and have thus revised the title accordingly. We now propose the title to be: *"Deletion of Trim28 in Committed Adipocytes Promotes Obesity but Preserves Glucose Tolerance"*.

This title specifically reflects the increased adiposity, whilst also acknowledging that glucose tolerance in these animals is not worsened by the obesity phenotype. It also avoids the term "metabolic health", which was the primary concern of the reviewer. Further to this change to the title, we have also made changes to the manuscript to acknowledge the reviewers interpretation of the data as being suggestive of adipose tissue metabolic inflexibility. See:

- Results: Section title on Page 11
- Results: general text on page 12, 25, 17 and 20
- Discussion: general text on page 21, 23 and 26

REVIEWERS' COMMENTS

Reviewer #2 (Remarks to the Author):

We thank the authors for their effort in trying to address our comments. We still have some points of disagreement regarding their answers to our comments as listed below.

Comment 1: We agree with the authors that the bright field images do not show a clear difference between the WT and shTRIM28 cells. However, the western blot now presented in Figure S8 instead shows an increase of C/EBPa expression in shTrim28, but it is not easy to conclude with an n=1, as for the brightfield images. Unfortunately, the authors could not provide gene expression analysis as asked that would have provided a better understanding of the role of Trim28 in adipogenesis and lipid storage/release in vitro.

Comment 2: We reiterate our point that the EE data is not convincing. The point of the ANCOVA is to test whether the two lines are parallel, therefore showing a shift, as detailed in PMID: 22205519. The EE phenotype is not critical for the manuscript message to our opinion but is somewhat confusing for the readership to claim an EE phenotype that seems unlikely.

Regarding the RER data, we are pleased that the authors emphasise more on this aspect, in our opinion, would be the core of their phenotype. We want to reinforce the point that, in addition to a substantial reduction of the RER in the KO mice (-10 in a scale going from 0.7 to 1 is not subtle), it is also the difference in between the maximum and nadir RER that relate to impaired metabolic flexibility. The reduced RER can reflect the incapacity of the AT to store lipids (besides the impaired release) therefore used as an energetic substrate in the organism.

The concept that the liver is compensating for the reduced AT lipolysis is of potential interest but purely speculative at this stage and not supported by the data. Accordingly, the reduced lipolysis in the AT would still result in a reduced FFA plasma level with an increased plasma TG level if the liver was secreting more VLDL, but plasma TG level are also lower in the KO. The reduced AT lipolysis in vivo remains unclear.

We acknowledge that the authors took on board our suggestion regarding the title change and appreciate their effort inappropriately discussing their findings.

Response to Reviewers:

We thank the reviewer for providing further insight into the potential underlying mechanisms in our interesting obesity model. We are pleased that the reviewer is appreciative of our efforts to answer the questions that have posed.

Reviewer #2 (Remarks to the Author):

We thank the authors for their effort in trying to address our comments. We still have some points of disagreement regarding their answers to our comments as listed below.

Please see our specific response below

Comment 1: We agree with the authors that the bright field images do not show a clear difference between the WT and shTRIM28 cells. However, the western blot now presented in Figure S8 instead shows an increase of C/EBPa expression in shTrim28, but it is not easy to conclude with an n=1, as for the brightfield images. Unfortunately, the authors could not provide gene expression analysis as asked that would have provided a better understanding of the role of Trim28 in adipogenesis and lipid storage/release in vitro.

The reviewer's prior concern was that the reduced lipolysis observed in Trim28 depleted 3T3-L1 cells may have been due to reduced differentiation (and therefore an intrinsic inability to undergo similar rates of lipolysis). The reviewer agrees that the bright field images, which often provide a broad insight into differentiation capacity, shown in Supplemental Figure 8 suggest that there is indeed no difference in differentiation. This supports our hypothesis that Trim28 depletion leads to specific reductions in lipolysis. Furthermore, we also provide Western blot data as further evidence of this, which as the reviewer points out, may actually show a slightly improved capacity to differentiate (potentially increased C/EBPa abundance). This of course is only supportive evidence and definitive proof would require more rigorous testing of this effect, which is not within the scope of the current manuscript.

Comment 2: We reiterate our point that the EE data is not convincing. The point of the ANCOVA is to test whether the two lines are parallel, therefore showing a shift, as detailed in PMID: 22205519. The EE phenotype is not critical for the manuscript message to our opinion but is somewhat confusing for the readership to claim an EE phenotype that seems unlikely.

We thank the reviewer for their continuing expertise on EE. We agree that the nuanced phenotype observed in the EE data is not a critical determinant in the manuscript, however it is necessary for understanding the broader context – hence why we included these data in our manuscript. We have revisited the text in the results section regarding EE, and have performed further minor edits to ensure the readers are not confused by our claims.

Regarding the RER data, we are pleased that the authors emphasise more on this aspect, in our opinion, would be the core of their phenotype. We want to reinforce the point that, in addition to a substantial reduction of the RER in the KO mice (-10 in a scale going from 0.7 to 1 is not subtle), it is also the difference in between the maximum and nadir RER that relate to impaired metabolic flexibility. The reduced RER can reflect the incapacity of the AT to store lipids (besides the impaired release) therefore used as an energetic substrate in the organism. The concept that the liver is compensating for the reduced AT lipolysis is of potential interest but purely speculative at this stage and not supported by the data. Accordingly, the reduced lipolysis in the AT would still result in a reduced FFA plasma level with an increased plasma TG level if the liver was secreting more VLDL, but plasma TG level are also lower in the KO. The reduced AT lipolysis in vivo remains unclear. We acknowledge that the authors took on board our suggestion regarding the title change and appreciate their effort inappropriately discussing their findings.

We have made minor edits to the manuscript results section to encompass the comments made here relating to RER and metabolic inflexibility.